



# Development of new strategies for optimized structural monitoring of wind farms: description of the experimental field

João Pacheco[1], Silvina Guimarães[2], Carlos Moutinho[1], Miguel Marques[2], José Carlos Matos[2], Filipe Magalhães[1]

[1]CONSTRUCT- VIBEST, Faculty of Engineering, University of Porto, Porto, 4200-465, Portugal
[2]INEGI, Porto, 4200-465, Portugal

*Correspondence to*: João Pacheco (ec11142@fe.up.pt)

**Abstract.** The main goal of the recently started WindFarmSHM research project is the development, validation and optimization of monitoring strategies to be applied at the level of the wind farm, which should be able to evaluate the structural
condition of a set of wind turbines and their consumed fatigue life, using the response to operation loads. In this context, a quite extensive experimental campaign is being performed in Tocha wind farm, an onshore wind farm located in Portugal, which includes the simultaneous instrumentation of several wind turbines adoting strain gages, clinometers and accelerometers distributed in the tower and blades. This paper introduces the Tocha wind farm, presents the different layouts adopted in the instrumentation of the wind turbines and shows some initial results from the already fully instrumented wind turbine. At this
preliminary stage, the capabilities of the very extensive monitoring layout will be demonstrated and it will be evaluated the ability of the different monitoring components to track the modal parameters of the system composed by tower and rotor.

## 1 Introduction

Wind energy is one of the most promising renewable energy sources, having registered a truly remarkable evolution in the last two decades. This evolution, practically worldwide, is verified both in terms of installed capacity, as well as in terms of
technological evolution. In the EU, wind was the fastest growing energy source between 2005 and 2017, surpassing coal in 2016 as the second largest total installed power generation capacity (EWEA, 2018). Future forecasts are equally optimistic. It is expected that cumulative capacity of wind energy in the EU will continue to grow and that it will even double in a minimum interval of 10 years, considering the most optimistic forecast (EWEA, 2017). Thus, based on this scenario, it is possible to identify several challenges that will arise in the coming years. Among them, the following stand out:

• Costs of energy production: the reduction of the unit cost of wind energy is a major factor to guarantee the competitiveness and growth of the wind sector. The reducing operation risk through monitoring is one away;

   • The extension of the lifespan of the existing wind turbines: wind turbines were designed to operate 20 years, so it is estimated that about one half of the accumulated capacity currently installed in the EU will reach the end of design





life in 2030 (EWEA, 2017). It is therefore essential to create a regulatory framework that defines the rules for the

actions to be taken when the expected design life of the structures is exhausted;

- Limited technical knowledge: the increase in the size of wind generators and the exploration of offshore sites still involve a certain degree of uncertainty.

Considering this background, the main goal of the WindFarmSHM research project is the development, validation and optimization of new methodologies to continuously assess the structural elements of wind turbines: tower, blades and

foundation. The monitoring strategy is being designed to be applied in the context of a wind farm, adequate for onshore and floating solutions, using optimized instrumentation layouts at a subgroup of wind turbines, and taking profit from the data provided by the acquisition systems already available in all wind turbines (SCADA), for the use of extrapolation techniques to assess all the wind turbines of the same wind farm (Figure 1).

The research project will include three monitoring layouts of wind turbines of an onshore wind farm, comprehending

accelerometers, strain gages and clinometers and the development of numerical models for the generation of artificial experimental data to validate the monitoring strategy in floating wind turbines.

The data processing will be based on the continuous evaluation of the parameters that drive the structure dynamic behaviour (vibration frequencies and damping) estimated from the structure response to ambient excitation (wind, waves, currents, soil vibrations) and advanced statistical modelling, having in mind two main goals: detection of stiffness reductions motivated by

the appearance of damage and evaluation of the remaining fatigue life of the main structural components (Figure 1).

In the project a very extensive instrumentation will be deployed in order evaluate different monitoring layout alternatives, but the final goal it to propose a minimal optimized monitoring layout based on a few number of easy to install sensors.

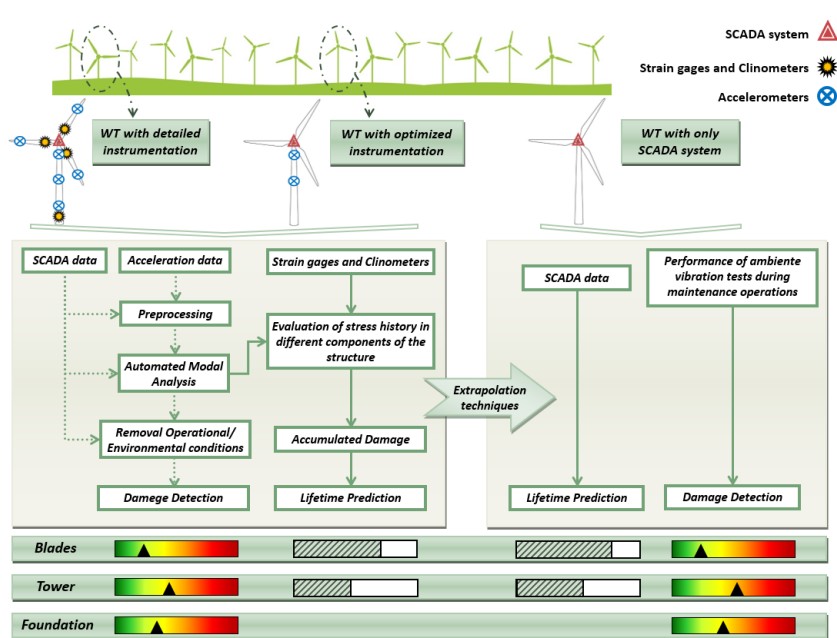

**Figure 1. Monitoring strategy.**




## 2 Tocha Wind Farm

The Tocha wind farm is owned by EDP Renewables and started its operation in May 2012. It is located in the central region of Portugal approximately 3 km from the coastline. It consists of five Vestas wind turbines, model V100 with 1.8 MW of rated power, totalling 9.0 MW of installed power. Figure 2 presents a geographic location of the wind farm and the distribution of the five wind turbines, identified with numbers that will be used throughout this work. This figure also identifies a substation position, as well as a meteorological mast.

It is important to note that the wind farm fits into a coastal area, with very soft orography of the terrain and where the foundation's soil is predominantly sandy, which is why deep foundations are used in all wind turbines. Thus, the steel tower of the turbines is connected to a 14-by-14 m concrete slab with variable height (1.50 m at the ends and 3.00 m in the central area). In turn, the slab is supported by sixteen concrete piles with 1 m diameter.

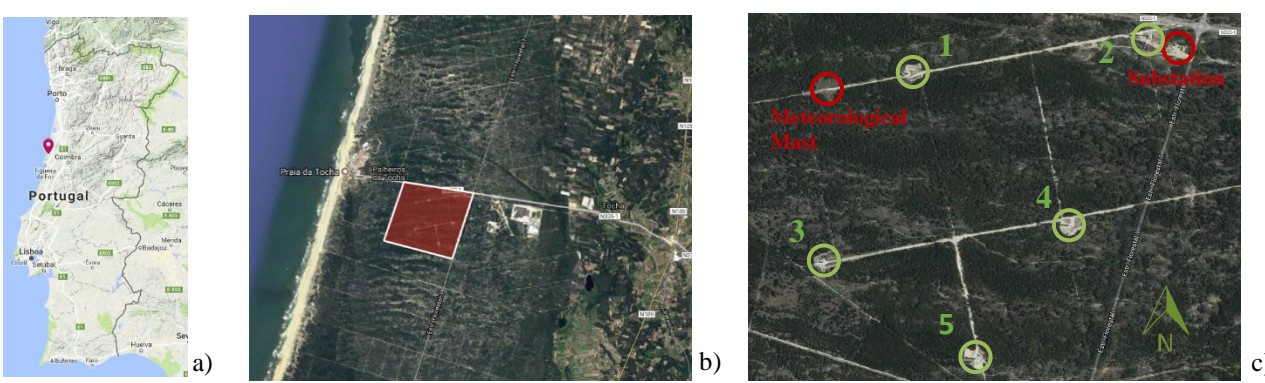

**Figure 2. Tocha wind farm: a) Geographic location in Portugal (© Google, n.d.-a); b) View of the implantation area (© Google, n.d.-b);** 60 **c) Identification of wind turbines and auxiliary structures (© Google, n.d.-c).**

Figure 3 shows a wind rose, which characterizes the wind speed and direction for the year 2017 at the Tocha wind farm. The predominant wind direction is approximately north. Thus, considering the very smooth terrain and the proximity of the coast, wind generators 1, 2 and 3 are exposed to slightly disturbed offshore winds, while the remaining generators are exposed to wind with additional turbulence caused by the wake effects.

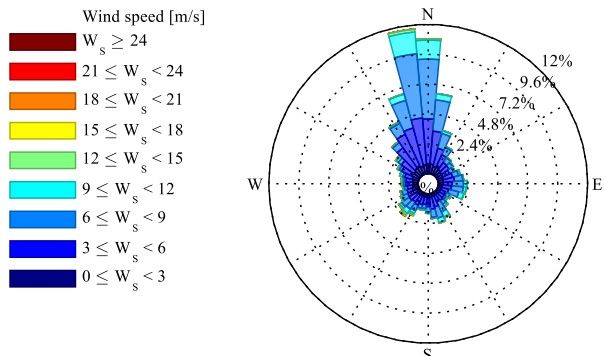

**Figure 3. Characterization of the wind conditions observed during 2017 (SCADA data of wind turbine 1).**



Vestas V100-1.8MW wind turbine is an onshore turbine model with a 100 m diameter rotor. It is a variable speed, 3 blades rotor with individual pitch control for each blade. The hub is placed at a height of 95m and is supported by a steel tower, of variable height in section, composed of four segments that are connected by boltedconnections. The wind turbines operate for wind speeds between 4 and 20 m/s and achieve the rated power for wind speeds of about 12m/s (Figure 4).

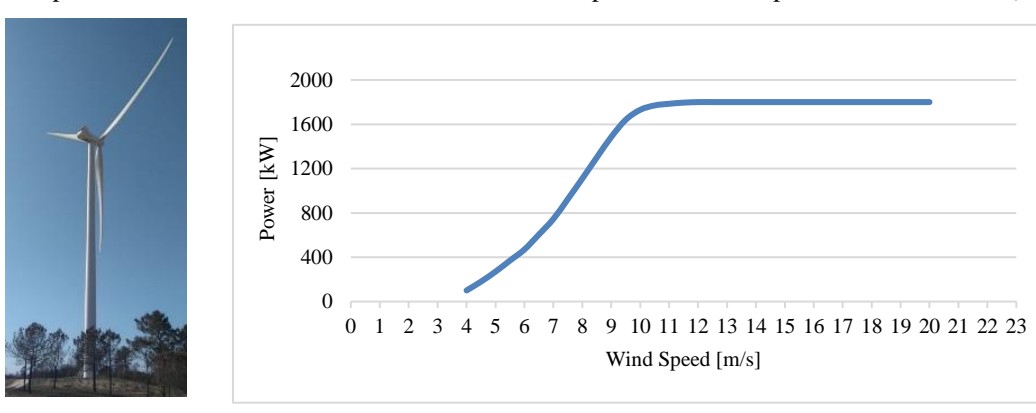

**Figure 4. Photo of one wind turbine at Tocha wind farm and power curve (https://en.wind-turbine-models.com).**

## 3 Preliminary evaluation of the modal properties of the wind turbines

In order to obtain an initial estimate of the wind turbine dynamic properties before the installation of the monitoring systems,
a set of ambient vibration tests was performed in four of the five wind turbines in operating and non-operating conditions. Additionally, a numerical model of the wind turbines was deployed. In the next sections the preliminary evaluation of the modal properties of the wind turbines will be described.

### 3.1 Numerical Models

In order to understand and interpret the experimental results, a numerical model of the wind turbine was developed using Robot structural analysis software (Autodesk, 2016), according to the technical drawings provided by the manufacturer. It is a simplified model, in which only a static component of the structure is modelled. Rotational movement of the rotor and all control systems are disregarded, since the main purpose is to simulate the dynamic behaviour of the tower under the test conditions presented in the following section.

Still, it's important to note that advanced models are currently being developed in FAST (Sprague, Jonkman et al., 2015) and some results are presented in the next section. This preliminary model has also important to obtain structural information then used in the FAST model (Pimenta, Branco et al., 2019).





### 3.2 Ambient Vibration Tests

The set of ambient vibration tests was divide into two campaign. In the first campaign, the main goal was to accurately identify the natural frequencies and the configuration of the tower vibration modes, considering two different situations: wind turbine in operating conditions and wind turbines in non-operating conditions (the rotor was stopped or idling). At this stage, only the wind turbine 1 was tested. The accelerations were measured with 4 standalone seismographs (Figure 5 a), with internal tri-axial force balance sensors, that were placed in the horizontal platforms of the tower (Figure 5 b).

In the second campaign, the main objective is to identify the natural frequencies of all wind turbines of the wind farm, in order to characterize the variability of the natural frequencies. The same equipment was used and with the same data acquisition parameters, but only the two highest sections of the towers were instrumented. It should be noted that in this second tests the rotor of the wind turbines was stopped.

The collected acceleration time series were first analysed in the frequency domain and then processed with Covariance driven

Stochastic Subspace Identification method (SSI-COV) (Magalhães and Cunha, 2011). The operating scenarios observed during the performance of the ambient vibration tests are shown in Figure 5 c) (1st campaign: red circles, 2nd campaign: green triangle). It can be seen that the wind conditions observed during the two test are quite different.

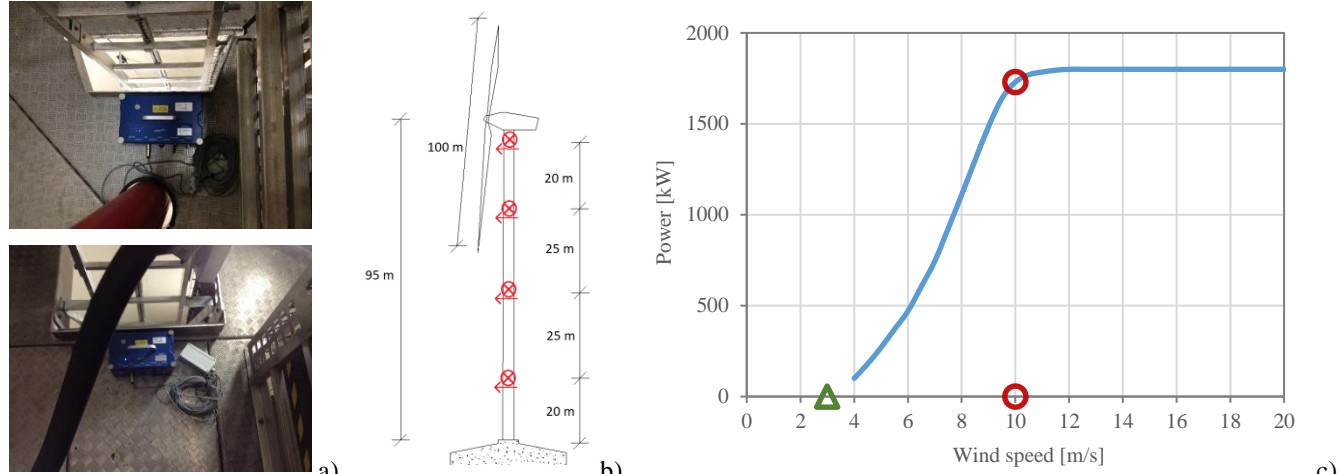

**Figure 5. Ambient vibration tests: a) Sensors; b) Section instrumented; c) Operating scenarios.**

Figure 6 shows some of the results obtained from the first campaign in non-operating condition. The top left plot shows two

averaged normalized spectra (ANPSD): one for the for-aft direction, FA, (perpendicular to the rotor plane) and another for the side-side direction, SS, (parallel to the rotor plane). It is possible to identify several abscissa in correspondence with the most relevant peaks of the spectrum, which represent good estimates of natural frequencies. Among the various peaks identified are two that clearly stand out: one near 0.25 Hz and another near 1.80 Hz. These peaks correspond to the first and second pairs of tower bending modes. In the stabilization diagram covering the full frequency range under analysis, the position of the first

three pairs of tower bending modes is marked. The two zooms presented at the bottom of the figure show that with the SSI-COV method it is possible to separate the very close modes within the first two pairs of frequencies. There are still other stable



pole alignments relevant to the dynamic characterization of the structure, however they are probably associated with vibration modes dominated by the rotor, which can only be identified and characterized using the more detailed instrumentation, which will be described in the next section.

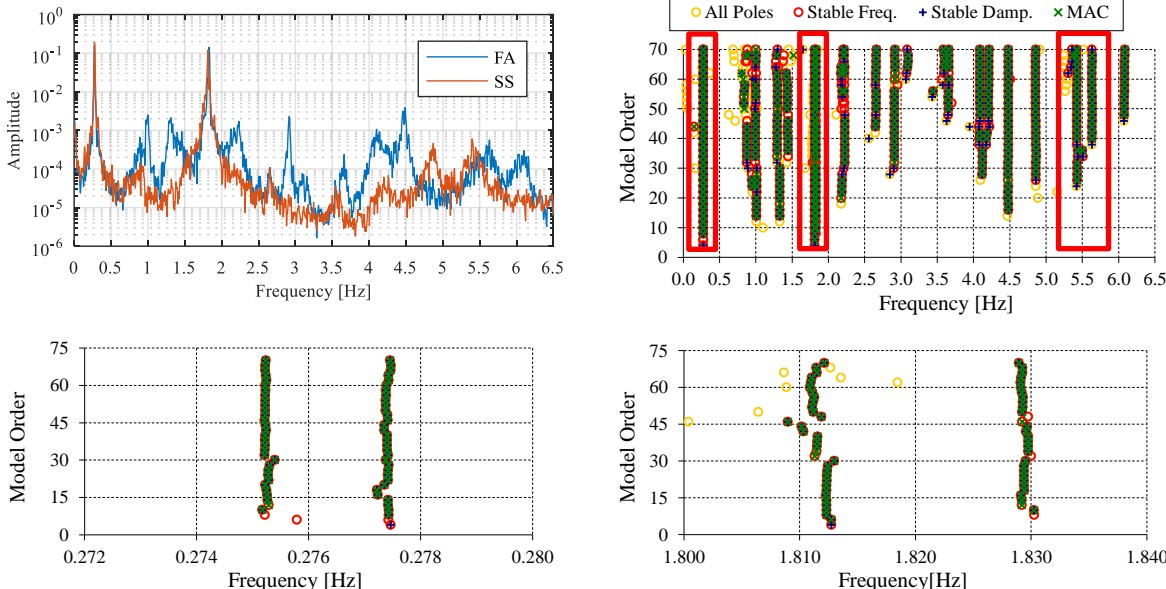

**Figure 6. Ambient Vibration test results for wind turbine 1: average power spectra for FA and SS directions, stabilization diagram produced by the SSI-COV method and two zooms of this diagram.**

Figure 7 presents the identified mode shapes and natural frequencies, which are compared to numerical results obtained from a preliminary simple numerical model. There is an excellent relationship between numerical results (blue line) and experimental results (red circles).

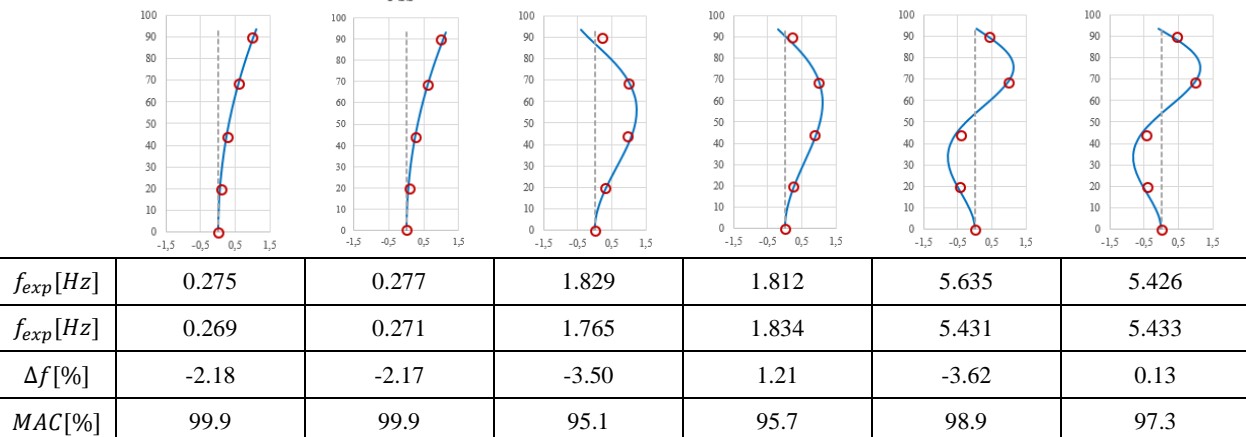

| | 1 FA | 1 SS | 2 FA | 2 SS | 3 FA | 3 SS |
|---|---|---|---|---|---|---|
| $f_{exp}[Hz]$ | 0.275 | 0.277 | 1.829 | 1.812 | 5.635 | 5.426 |
| $f_{exp}[Hz]$ | 0.269 | 0.271 | 1.765 | 1.834 | 5.431 | 5.433 |
| $\Delta f[\%]$ | -2.18 | -2.17 | -3.50 | 1.21 | -3.62 | 0.13 |
| $MAC[\%]$ | 99.9 | 99.9 | 95.1 | 95.7 | 98.9 | 97.3 |

**Figure 7. Mode shapes and natural frequencies identified with the ambient vibration test and numerical model of wind turbine 1.**





Still, for the first test campaign, it is important to understand the influence that the normal operation of the rotor has on the dynamic characteristics of the structure. Thus, Figure 8 compares ANPSD obtained for stopped rotor and in operation. It should be noted that the represented ANPSD were calculated considering together the signals measured along the FA and SS directions. The dashed vertical lines represent the harmonic frequencies associated with to rotor operation. The results obtained

in terms of natural frequencies ($f$) are also compared for the identified vibration modes for the two analysed situations.

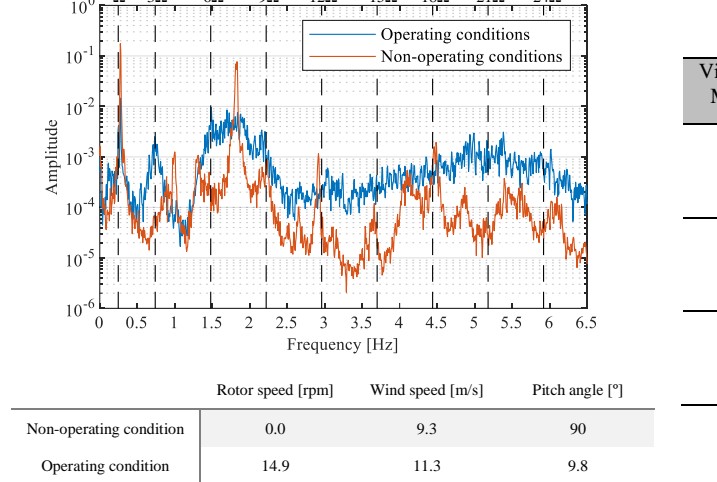

| Vibration Modes | Non-operating condition $f$ [Hz] | Operating condition $f$ [Hz] |
|---|---|---|
| 1 FA | 0.275 | 0.296 |
| 1 SS | 0.277 | 0.275 |
| 2 FA | 1.829 | 1.795 |
| 2 SS | 1.812 | 1.896 |
| 3 FA | 5.635 | 5.489 |
| 3 SS | 5.426 | 5.355 |

| | Rotor speed [rpm] | Wind speed [m/s] | Pitch angle [°] |
|---|---|---|---|
| Non-operating condition | 0.0 | 9.3 | 90 |
| Operating condition | 14.9 | 11.3 | 9.8 |

**Figure 8. Ambient Vibration test results for wind turbine 1 in operating and non-operating conditions: average power spectra and natural frequencies.**

Figure 9 compares the results obtained for the four tested wind turbines. All of them present quite similar natural frequencies, but wind turbine 5 seams to present a slightly different behaviour. For this reason and because this is the wind turbine where

higher turbulence is expected the monitoring camping will be focused on wind turbines 1 and 5.

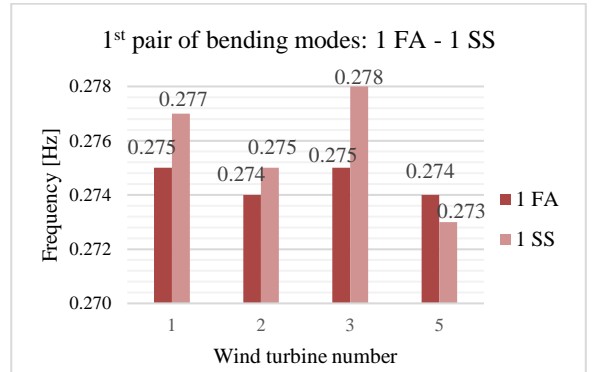

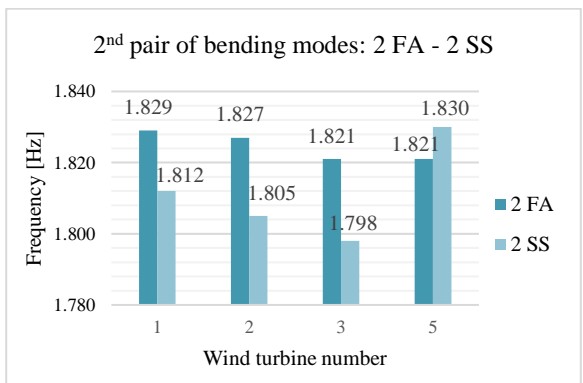

**Figure 9. Comparison of the natural frequencies of four wind turbines (1ˢᵗ and 2ⁿᵈ pairs of bending modes).**



## 4 Monitoring Systems and Preliminary Monitoring Results

The experimental campaign in Tocha wind farm involves the simultaneous monitoring of several wind turbines during a period
of about two years. In order to obtain data representative of the dynamic behavior of all generators and based on the results of
the ambient vibration tests described above, the experimental campaign includes the following three instrumentation layouts:

- o   A very complete monitoring layout installed on wind turbine 1;
- o   An intermediate monitoring layout installed on wind turbine 5;
- o   A simple monitoring layout to be installed on the other wind turbines, considering shorter instrumentation periods;

Based on the wind conditions of the site (Figure 3) and the position of each wind turbine in the wind farm (Figure 2) wind
turbine 5 is the wind turbine where higher turbulence is expected because it is in the wake of the other wind turbines, while
wind turbine 1 is exposed to less disturbed winds. For this reason, wind turbine 1 was instrumented according to the complete
layout, while the intermediate layout was applied in wind turbine 5.

The main goal of simple monitoring layout is to characterize the differences of the behaviour wind turbines and to understand
the interaction between neighboring wind turbines. This simple layout will be applied to all other structures, considering time
periods limited to two or three months. If justified, or atypical behaviors are identified, these generators can be instrumented
by adopting a complementary layout, suitable for each situation that is intended to be analyzed.

The complete monitoring layout includes the following components:

- o   Two alternative systems to characterize accelerations at the tower: a commercial system based on a set of very low
noise accelerometers and a customized low-cost system based on MEM accelerometers designed and assembled in
        FEUP (Moutinho and Cunha, 2019);
- o   Strain gages at the tower base to characterize the stresses during different operating conditions;
- o   Clinometers to characterize the rotations and, indirectly, the bending moments at the base of the tower;
- o   A set of fiber optic strain gages to estimate bending moments at the blades roots;
o   MEM accelerometers placed at the blades (10 meters from the root) to characterize their dynamics.

The intermediate layout includes the characterization of the accelerations at the tower through the optimized low-cost MEM
accelerometer system and the characterization of the rotor dynamic behavior through fiber optic strain gauges and MEM
accelerometers installed on the blades.

Finally, the simple monitoring layout consists solely of using the MEM accelerometer system to collect data regarding tower
vibrations. As mentioned, this system will be applied to all other wind turbines and may be supplemented as appropriate.

It should be noted that data on the environmental and operational conditions of each generator is being  obtained through the
SCADA system. The meteorological mast is also important to characterizethe history of environmental conditions in the park
(wind direction and wind speed) since the beginning of its operation. This information is very useful for estimating the current
state of fatigue of the various structures.





Wind turbines 1 and 5 are already instrumented. The following section describes the various instrumentation systems adopted, together with the presentation and analysis of preliminary results for wind turbine 1. Since the installation of these components is still being adjusted and the amount of data acquired is still limited, the results presented here are intended to demonstrate what is being measured, to certify the correct functioning of the systems and to demonstrate the capabilities of the most complete monitoring layout.


### 4.1 Tower Monitoring System: Accelerometers

In order to obtain the best possible characterization of the tower accelerations a commercial system based on 6 force-balance accelerometers connected to a 24bits acquisition system was deployed. As depicted in Figure 10, this involved the instrumentation of 3 sections of the tower along two orthogonal horizontal directions. The sensors are connected by cables to

a central acquisition system that continuously records acceleration time series. This data is accessible from FEUP through an internet connection.

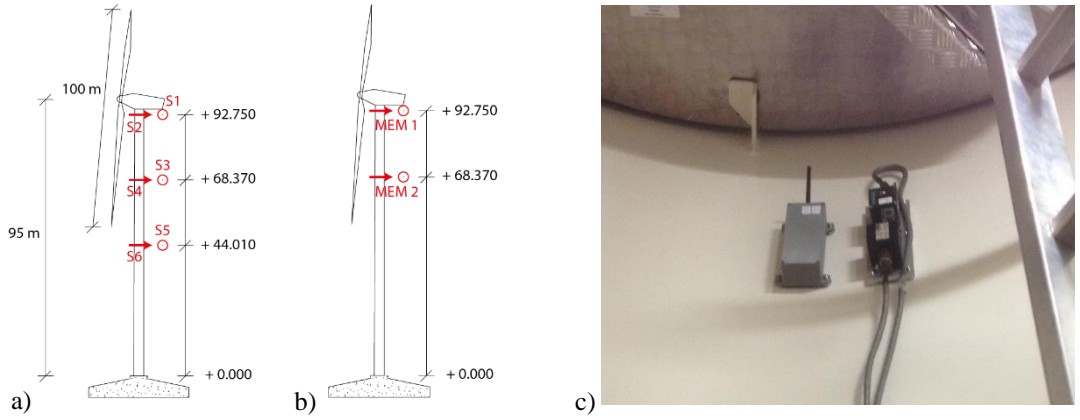

**Figure 10. Sections instrumented with accelerometers: a) force balance sensors; b) MEM sensors; c) photos of the force balance sensors (connected to cables) and of the MEM based acquisition system (grey box with antenna).**

Complementary, a MEM based system was also installed. This is a standalone system developed at FEUP that integrates a tri-

axial acceleration sensor (in this application just the two horizontal directions are being recorded), a set of batteries that ensure 5 months of continuous operation, a memory card for data storage, high-precision clocks and a radio for data transmission (in the present application the data transmission is limited to state-of-health parameters to increase the system autonomy). Two of these devices were installed in the tower in the positions marked in Figure 10. One of the project goals is the development and test of easy to deploy and cost effective systems for wind turbines testing and monitoring, so the evaluation of the performance

of these devices designed and assembled in FEUP is very relevant.

Figure 11 shows two examples of spectra obtained from acceleration series recorded by the two alternative sensor under test, considering the wind turbine in production (figure on the right) and parked (figure on the left). It appears that the system designed at FEUP demonstrates a performance that is comparable to the more expensive and difficult to install commercial





system (KMI). These figures are in accordance with the results of the ambient vibration test presented above. Under non-
operating conditions, the peak pairs associated with the first two tower mode pairs clearly stand out. In operating conditions,
additional peaks associated with the rotor rotation frequency appear. The peaks associated with the second pair of bending
modes become much more diffuse, which  makes their tracking over time quite challenging.

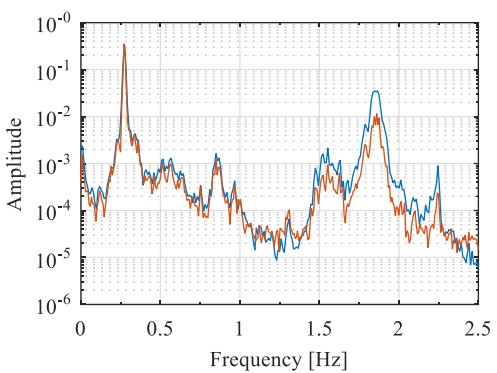
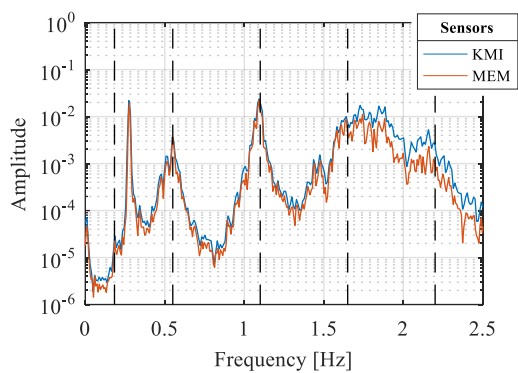

Pitch angle = 90º; Rotor speed = 0.0 rpm; Wind speed = 2.9 m/s          Pitch angle = -1.7º; Rotor speed = 11.1 rpm; Wind speed = 5.4 m/s

**Figure 11. Power spectra from force balance (KMI) and MEM sensors (1Ω, 3Ω and 6Ω marked with dashed vertical lines).**

Figure 12 shows the colormaps obtained from spectra of singular values calculated with the acceleration time series acquired
with the commercial system during January 2019, after their projection according to the FA and SS directions. As might be
expected, variations in frequency content are observed over time due to varying operating conditions. It is also possible to
visually track the time evolution of the natural frequencies associated with the first two pairs of tower modes.

The data collected by both systems is being processed with the algorithms presented in (Oliveira, Magalhães et al., 2018).

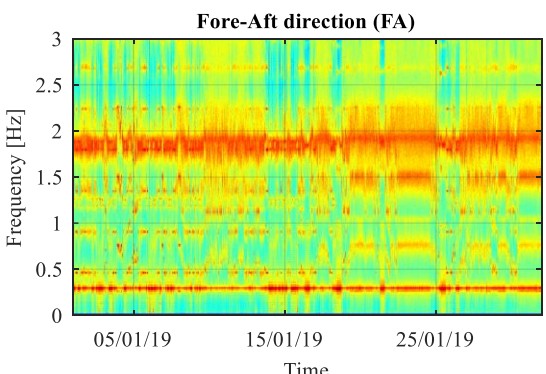
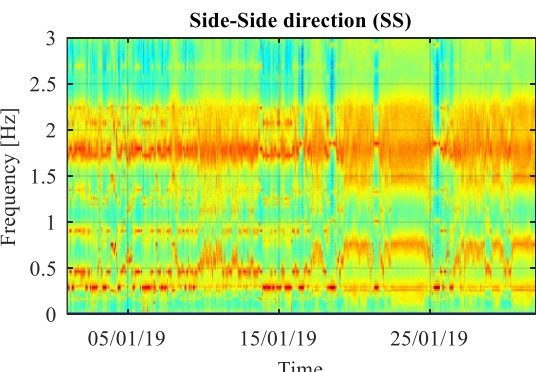

**Figure 12. Colour maps with singular value spectra for the FA and SS directions during January/2019.**




## 4.2 Tower Monitoring System: Strains and Rotations

These monitoring components are essential for fatigue assessment of the tower and on important  goal is the evaluation of two
alternatives for estimating static and dynamic bending moment diagrams along the tower: using extension measurements and
rotation measurements, combined with accelerometers.

The strains system is composed of six 2D rosette strain gages (measurement of the strain in two orthogonal directions) and 4
temperature sensors. In order to try to evaluate the static bending moment diagrams evolution along the tower, the six strain
gauges are distributed in two sections: four sensors at 6.5m from the base of the tower (bottom section) and two sensors at
7.7m (top section) as shown in Figure 13. The four temperature sensors are located in the bottom section, close to the strain
gauges. Measuring deformation in the direction perpendicular to the tower axis and temperatures is important to allow the
evaluation of alternative procedures to minimize the influence of temperature on the measured longitudinal deformations.

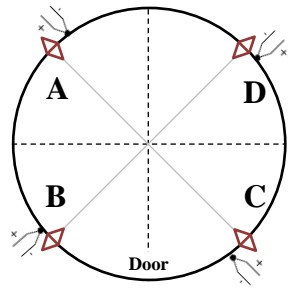
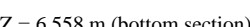
Z = 6.558 m (bottom section)

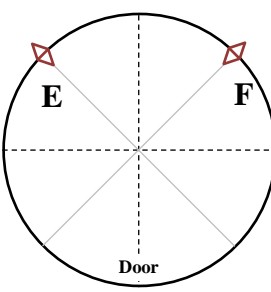
Z = 7.758 m (top section)

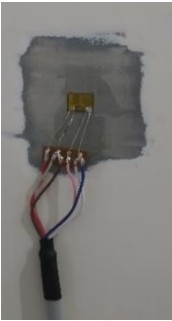

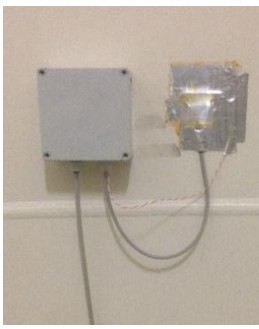

**Figure 13. Locations of the strain gages (◊) and temperature sensors ( ); photo of a 2D rosette strain gage before protection; and photo of the strain rosette and temperature sensor after protection and box for signal conditioning.**

The installation of the clinometers aims to measure the rotation at the base of the tower and to alternatively estimate the
extensions from the measurement of rotations in two close sections. The main advantage of estimating bending moments from
rotations is that the installation of the clinometers is less intrusive than the installation of strain gauges, which involves
removing of tower painting. The three clinometers were installed along the vertical alignment formed by strain gauges A and
E. One of the clinometers was installed close to the foundation (near the base flange), while the two ones are positioned
according to the diagram in Figure 14.





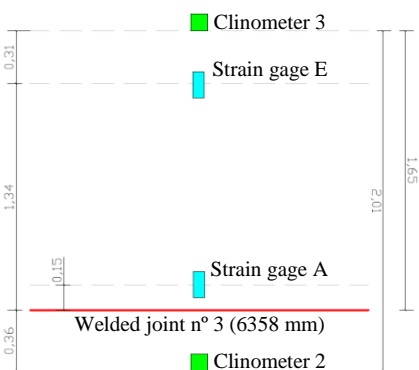
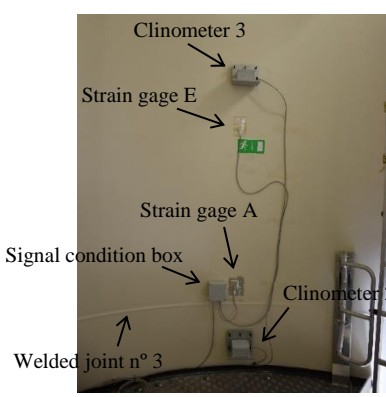

**Figure 14. Elevation diagram and photograph of the position of the clinometers and strain gauges along the same vertical alignment.**

The two monitoring components are connected to a National Instruments digitizer and processor (model cRio 9056 - http://www.ni.com), installed at the base of the tower. Data acquisition is ensured by a program developed in LabView for this

specific application.

Figure 15 a) shows an example of the strains time series obtained for stop event of the rotor. Although the main purpose of this monitoring system is to characterize the static component of the response, it is possible to characterize the dynamic component with good accuracy. With this data, it will be relevant to test and compare the various approaches for estimating the dynamic stresses in the tower from acceleration measurements (Maes, Iliopoulos et al., 2016).

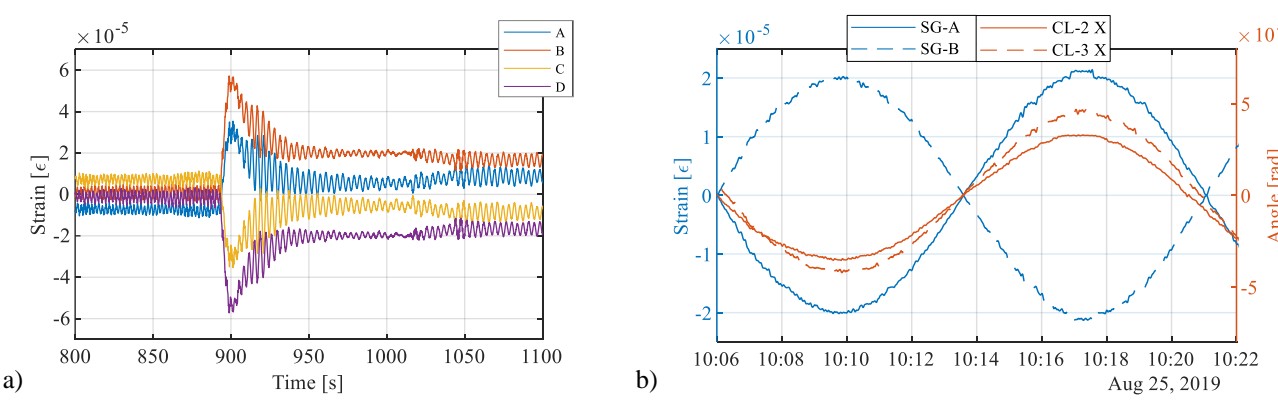

**Figure 15. Tower Monitoring System Strains and Rotations: a) Example of strain time series; b) Strain and rotations during nacelle yawing, SG – strains; CL - rotations (see sensor positions in Figure 13 and Figure 14).**

The records obtained from strain gauges are influenced by several factors, including the effect of temperature. Thus, the experimental determination of bending moments in the tower requires the acquired raw data to be pre-processed to obtain the real deformation. In the present application, as a first trial, the methodology presented in (Loraux, 2018) is being followed. In

a general way, this methodology consists of the following three steps: a) correction of the effect of temperature on strain gauges; b) signal correction based on the average value of the extensions recorded on diametrically opposed sensors; c) signal calibration according to (IEC 61400-13, 2015). For this last step it is necessary to have a record of strain time series measured during a 360º nacelle rotation, with wind speeds  lower than the generator cut-in wind  speed. The eccentricity of the nacelle




and rotor mass generates a sinusoidal signal in the sensors, being the mean value of this signal the zero baseline Figure 15 b).

Applying the described method to the recorded series, in Figure 16 the temporal evolutions of the bending moments observed in the bottom instrumented section are presented, for the two main directions, considering two alternative turbine operation scenarios. The experimental results are compared with numerical ones, obtained from a model developed on FAST and calibrated using the methodology described in (Pimenta, Branco et al., 2019).

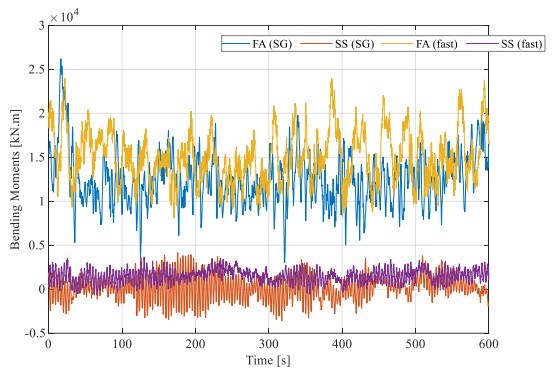 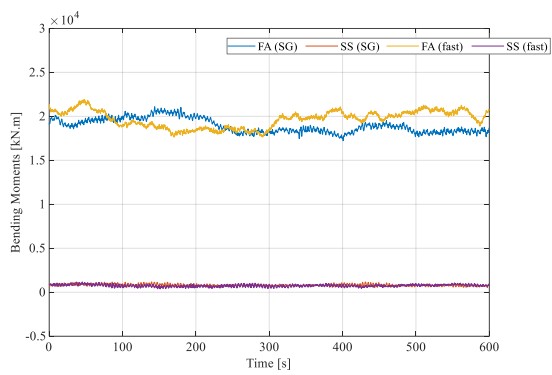

Rotor speed = 14.9 rpm; Wind speed = 13.0 m/s; TI = 14.6 %          Rotor speed = 13.0 rpm; Wind speed = 7.9 m/s; TI = 4.4 %

**Figure 16. FA and SS bending moments in the bottom instrumented section considering two different operating situations and**
**comparison with FAST numerical results (TI: turbulence intensity).**

Figure 17 shows the average spectra of longitudinal deformations recorded by sensors A, B, C and D (first row) and clinometer 3 (second row), considering the rotor parked (left) and in operation (right). These spectra show excellent agreement of results between the alternative monitoring components and demonstrate that it is possible to perform operational modal analysis from the data collected by all these systems.

Comparing the spectra with those shown in Figure 11, it is clear that the peaks corresponding to the tower bending modes are more pronounced and clearer, so measuring extensions can be very useful in distinguishing tower modes from the rotor modes observed in the tower.



**Figure 17. Averaged power spectra: longitudinal strains A to D (first row) and rotation at clinometer 3 (second row) in non-operating condition (left column) and operating condition (right column).**






### 4.3 Rotor Monitoring System: Accelerometers

The goal of this monitoring system is the characterization of the rotor under different operating conditions. The analysis of the results of the ambient vibration tests show the existence of several resonance frequencies that could not be attributed to the

tower fundamental modes. These are certainly related to  modes more dominated by the rotor. In addition, direct identification of rotor modes may be beneficial for automatically detecting blade changes, driven either by reduced stiffness due to damage or by additional masses due to ice formation. In this way, the same MEM based devices that were installed in the tower were also installed inside the blades, one in each blade, 10 m from the blade root, as shown in Figure 18.

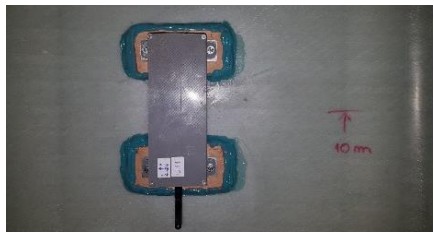
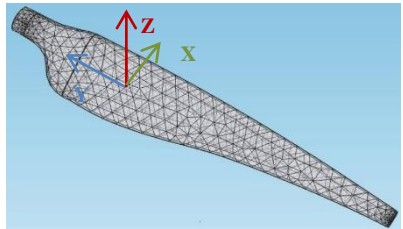

**Figure 18. MEM based system installed inside one blade and direction of measurements (X approximately aligned with edgewise**
**direction).**

From the simultaneous recording of the acceleration time series on the blades it is possible to estimate the modal parameters of the rotor, in particular their modal configurations. However, as this is a preliminary step, and since the data available so far is limited, only examples of the time series (Figure 19) and their spectra (Figure 20), considering the stopped rotor (left) and in operation (right) are presented. Signals $X$, $Y$ and $Z$ are in accordance with the referential presented in  Figure 18.

Considering the figures obtained with the rotor parked, in addition to the various peaks corresponding to the main tower modes, peaks are also identified for various other resonant frequencies that are certainly associated with the rotor modes. Already when the rotor is in operation, the adopted sensors measure the gravity, being the registered accelerations dominated by the rotor rotation frequency. Several other frequencies associated with vibration modes in flapwise ($Z$) and edgewise ($X$) directions can still be observed.

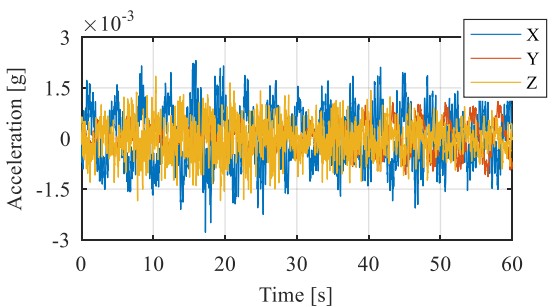
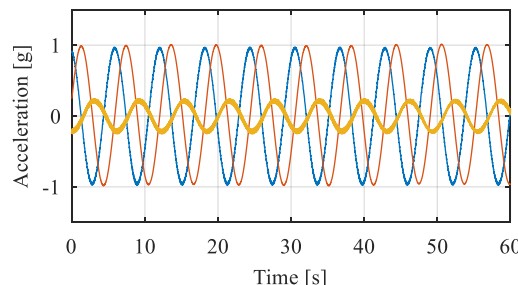

Pitch angle = 90º; Rotor speed = 0.0 rpm; Wind speed = 5.5 m/s        Pitch angle = 0º; Rotor speed = 10 rpm; Wind speed = 5.9 m/s

**Figure 19. Example of acceleration time series and corresponding operation parameters.**





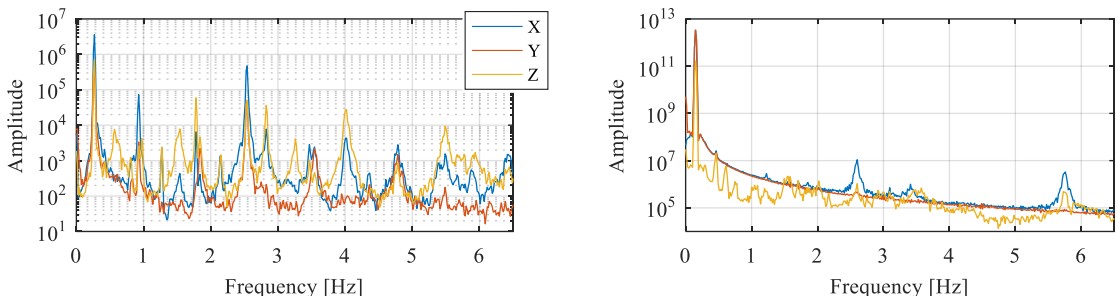

**Figure 20. Power spectra of acceleration time series represented in Figure 19.**

### 4.4 Rotor Monitoring System: Strains

The main goal of the blades strains monitoring is to collect data to estimate the fatigue condition of these elements, as well as
to evaluate their structural performance from the evolution of the continuously estimated modal parameters. On the other hand,
the joint analysis of the wind characteristics,  the moments acting at the blades and the bending moments at the tower will also
be relevant to understand the mechanism of transmission of loads from the rotor to the tower and to validate numerical
modelling.

The solution adopted is based on a commercial system provided by HBM / FiberSensing called WindMeter
(https://www.hbm.com). Each blade is instrumented with 4 fiber optic strain sensors and temperature sensors for compensation
of the temperature effects. As shown in Figure 21, each set of sensor is connected to a central acquisition system installed on
the hub, which in turn allows remote access to data via a 3G modem.

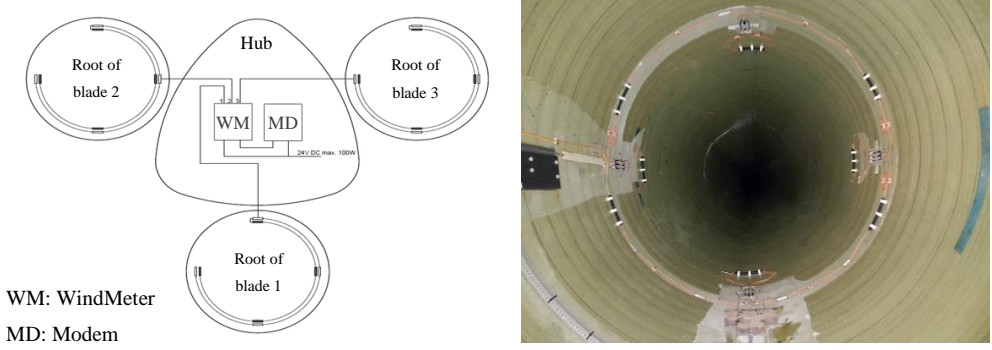

**Figure 21. Strain measurement at the blades root: wiring and photo.**

As an example, Figure 22 shows two strains time series and in the Figure 23 their spectra, considering the rotor stopped (left)
and in operation (right). Sensors $S1$ and $S3$ correspond to blade bending according to edgewise direction, while sensors $S2$ and
$S4$ correspond to flapwise direction. The following results show that the acquired data, besides being fundamental to obtain
the stress history for fatigue analyses, can also be used for operational modal analysis of the structure.





It should be noted that the deformations measured on the blades are not as sensitive to the tower bending modes as in the case of accelerations, since although tower movement produces blade movements, it does not lead to relevant bending levels. Thus, the spectra peaks shown in Figure 23 can only be motivated by the contribution of the blades modes.

By comparing the spectra of Figure 20 and Figure 23 for the parked situation, it is possible to identify several coincident peaks for the same frequencies. While in operation, the observed resonant frequencies depend on the rotor speed of the rotor, so the peaks do not coincide.

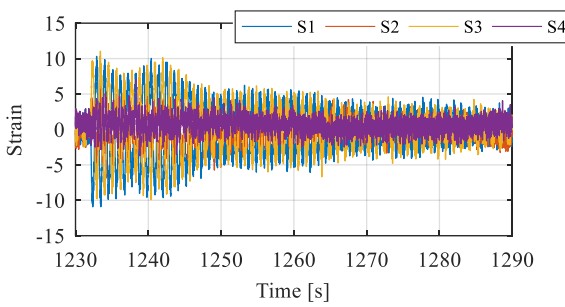
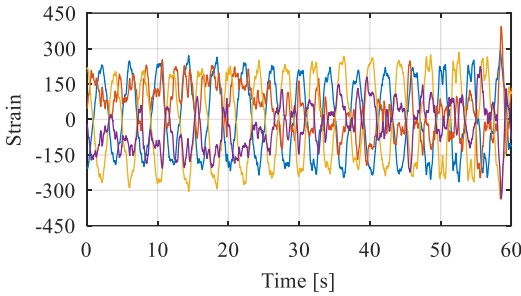

Pitch angle = 90°; Rotor speed = 0 rpm; Wind speed = 3.2 m/s     Pitch angle = 3°, Rotor speed = 15 rpm; Wind speed = 10.5 m/s

**Figure 22. Example of detrended strain time series (static component was removed) and corresponding operation parameters (S1, S3 bending in the edgewise direction; S2, S4 bending in the flapwise direction)**

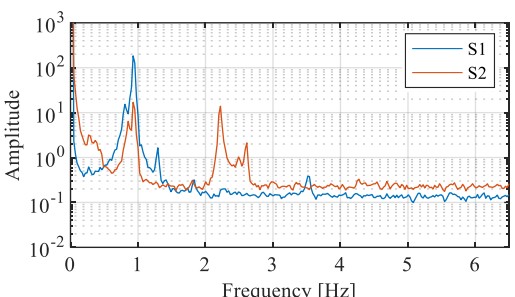
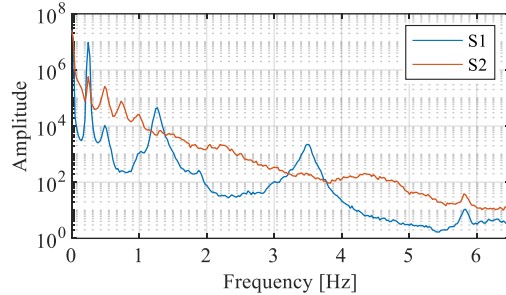

**Figure 23. Power spectra of the time series presented in Figure 22.**

As noted with respect to measuring tower extensions, a similar methodology was also followed for processing the blade strains records. Note that the calibration step according to the standard (IEC 61400-13, 2015) is not yet fully tunned. However, the data acquired so far allowed the elaboration of Figure 24, which represents the evolution of the bending moments at blade root B (wind turbine 1) to the flapwise and lead-lag directions as a function of wind speed and considering different turbulence intensities. Firstly, the moment value increases as the wind speed increases. When the wind turbine's nominal wind speed (9 $m/s$) is reached, the actuation of the pitch angle mechanism causes the momentum to decrease even though the wind speed continues to increase.

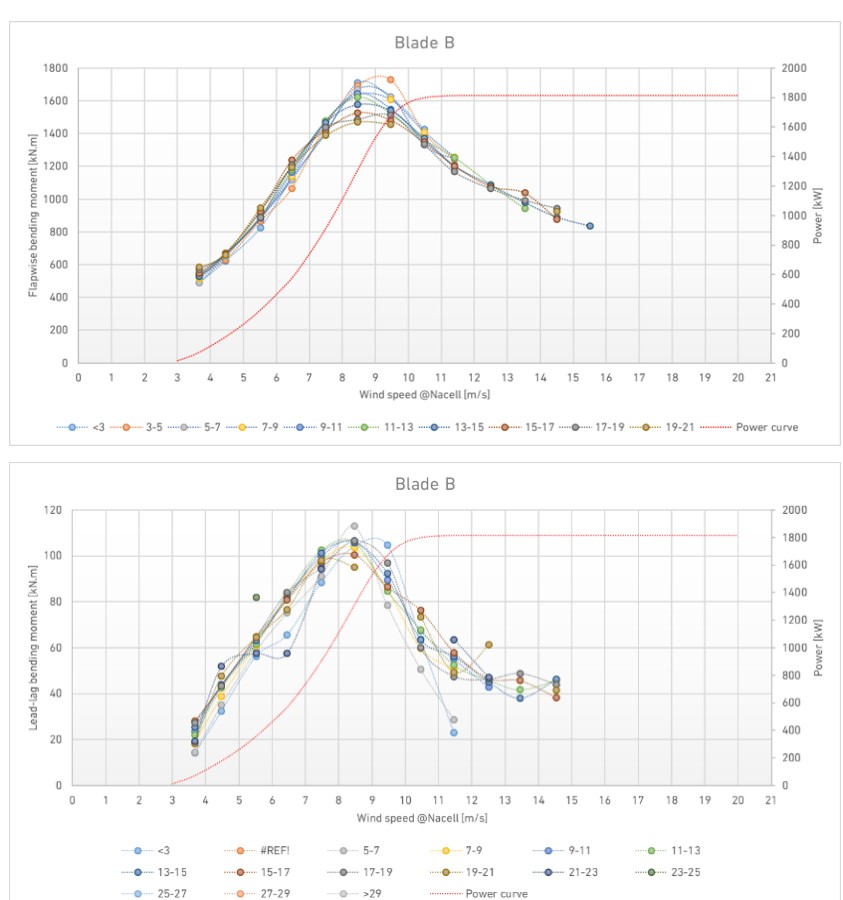

**Figure 24. Wind turbine power curve and bending moments recorded at blade root B according to flapwise direction (first) and lead-lag direction (second) as a function of wind speed and considering different turbulence intensities.**

## 5 Conclusions

This paper presented the quite extensive monitoring camping that is being conducted in Tocha wind farm, described the

installation of the monitoring components that are already in operation and presented some preliminary results.

The preliminary analyses performed in the frequency domain show that operational modal analysis has the potential to extract useful information from both strain and acceleration measurements performed either in the tower or in the blades.

A deeper processing of the data that is being continuously collected by all the monitoring components will certainly contribute to better understand the in-operation dynamic behavior of these quite complex structures, to devise processing procedures for

effective evaluation of their structure health and to calculate accumulated damage due to fatigue. This step will be instrumental in defining the most effective procedures for assessing structural performance and for estimating accumulated fatigue damage.

The analysis of data simultaneously collected in several wind turbines will be very important for understanding the relation between the observed fatigue wear and to devise techniques to extrapolate results from ones to the others.



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
