# Peer review of "Development of new strategies for optimized structural monitoring of wind farms: description of the experimental field"

_Wind Energy Science, 2020_

## Short Comment (SC1) · 2 Apr 2020

Dmitri Tcherniak

dtcherniak@bksv.com

p.2, l.48: "a few number of easy to install sensors" - " a few number" does not read good. p.4 the phrase "of variable height in section" is unclear same sentence: "bolted-connections" - mistype Please check the consistency using a space between the value and the unit: "12m/s" but "20 m/s", also "100 m diameter rotor" but "height of 95m" p.4, l.83: "since the main purpose is to simulate the dynamic behaviour of the tower". The main purpose of what? Of the entire project or of the numerical model? If the authors mean the entire project, it would be necessary to reflect this somewhere in the introduction. p.7, l.124 "the harmonic frequencies associated with to rotor operation" ->

two rotor operations? p.7, fig.8a: It is hard to see if there are "blue" peaks behind the red ones for the 1st and 2nd tower modes p.8, l.150: what is a MEM accelerometer? Do you mean MEMS accelerometers? p.16, fig.20. What are the units of the vertical axes?

———————————————

---

## Author Comment (AC1) · 6 Apr 2020

The authors thank the comments. We have considered the feedback in the revised manuscript. Each comment is further addressed below.

- Comment 1: p.2, l.48: "a few number of easy to install sensors" - " a few number" does not read good. - Response: In the project a very extensive instrumentation will be deployed in order evaluate different monitoring layout alternatives, but the final goal it to propose a minimal optimized monitoring layout based on reduced number of sensors that can be easily installed.

- Comment 2: p.4 the phrase "of variable height in section" is unclear same sentence: "boltedconnections" - mistype Please check the consistency using a space between the value and the unit: "12m/s" but "20 m/s", also "100 m diameter rotor" but "height of 95m" - Response: The hub is placed at a height of 95 m and is supported by a steel tower, with a hollow circular cross-section with variable diameter and thickness, composed of four segments that are connected by bolted connections. In the revised document, it will be consistently used a space between the value and the unit: 12 m/s, 95 m.

- Comment 3: p.4, l.83: "since the main purpose is to simulate the dynamic behaviour of the tower". The main purpose of what? Of the entire project or of the numerical model? If the authors mean the entire project, it would be necessary to reflect this somewhere in the introduction. - Response: The main purpose of the numerical model.

- Comment 4: p.7, l.124 "the harmonic frequencies associated with to rotor operation" -> two rotor operations? - Response: "the harmonic frequencies associated with the rotor operation (W, 3W, 6W ,...)"

- Comment 5: p.7, fig.8a: It is hard to see if there are "blue" peaks behind the red ones for the 1st and 2nd tower modes - Response: There is a blue peak for the 1st tower mode, but for the 2nd one there isn't. Under non-operating conditions, the peak pairs associated with the first two tower mode pairs clearly stand out. In operating conditions, additional peaks associated with the rotor rotation frequency appear. The peaks associated with the second pair of bending modes become much more diffuse, which makes their tracking over time quite challenging.

- Comment 6: p.8, l.150: what is a MEM accelerometer? Do you mean MEMS accelerometers? - Response: It is a typo that was repeated several times, we meant MEMS (micro electromechanical systems). This will be corrected in the revised manuscript.

- Comment 7: p.16, fig.20. What are the units of the vertical axes? - Response: These

are normalized power spectra, so without units.

---

## Referee Comment (RC1) · Anonymous Referee #1 · 29 Apr 2020

This paper presents the Tocha wind farm as well as the sensors installed and some initial results from those sensors. The paper is interesting, and it is useful to see the different types of results from different types of sensor. The purpose is as a precursor to future work, but I think the paper is interesting enough on its own merit. The paper is generally well structured and well put together, however I have several observations.

Individual questions and issues:

Section 3.1 needs more detail on the simple model as results are presented later. Some more specifics on thing such as how elements are modeled, what boundary conditions are used and what assumptions are made would be useful.

[Figure]

It's not clear how much data was used to construct the frequency tables in figure 8. Are these single observations of frequency or are they averages of multiple observations? It would also be good to know how much deviation is observed to give context to the level of difference between 'non-operational' and 'operational'.

Line 129: it's mentioned that turbine 5 behaves 'differently' to the other turbines. Although a difference can be seen in figure 9 it would be clearer for the reader to say in the text what this difference is.

In section 4.2, I don't think it's ever mentioned what the sampling rate of any of the sensors are. Since a comparison is generally invited between the different sensors, such as that strain gauges can be used for an OMA purpose, it would be useful for the reader to know how comparable these sensors are regarding aspects such as the sampling rates.

Line 242: It's mentioned that a model is conducted in FAST, also mentioned before in section 3.1, and the results are compared to the measurements. This is all moved on from too quickly. Why is a FAST model used? How important for that purpose is the difference from the measured results? Please give a bit more of the pertinent details on this and explain the aspects of this which might be of interest to the reader.

Technical corrections:

Figure 1 is good for expressing the process, but the bottom three rows are confusing, what do the bars to the right of 'Blades', 'Tower' and 'Foundation' mean?

Figure 9, it would help the reader to state in the caption that these measurements were from the non-operational condition.

The description in the text of figure 17 (line 246) doesn't quite match the figure. It seems the results for 'Force-balance accelerometers' was added without updating the text.

Line 122: please define the acronym ANPSD before using it.

A few minor typos and some grammatical errors throughout, though not too bad. Some examples:

I think you should capitalize 'Robot Structural Model'

Check grammar in line 86-87.

Table in Figure 7: I think the second fexp should be fmodel

Please check the grammar in the sentence at line 144.

---

## Referee Comment (RC2) · Lisa Ziegler (Referee) · 5 May 2020

I will specify my general comments below. Please find my detail comments (line-specific) in the pdf attached.

The authors present their experimental field for structural monitoring of onshore wind turbines. They introduce sensor setups and first results on modal parameters. The topic has high relevance for the wind industry due to aging fleet of assets.

The paper is cleary written, the content is sound and well presented.

Introduction misses a review on state-of-art and existing literature. What is the research

gap you wish to fill?

Presented results are clearly, however, I miss novelty here. Furthermore, I wish there would be critical discussion in the paper. For example, interesting questions would be: * Why is the specific instrumentation chosen? * How are number and positions of sensors chosen, e.g. sensitivity study of desired results to sensor palcement? * How do you deal with measurement noise and varying operational conditions? * How do you clear and pre-process data? In addition the following is missing or must be adapted: * Results on the comparison between bending moments obtained from strain gauges and clinometers shall be presented. * A feedback from results of blade monitoring to tower monitoring. Can you now explain some more of the excitation frequencies? * Blade results are presented although the calibration is not completed. Please finish first the calibration, then present results.

I do not understand why the results from FAST are presented. There are not enough details given to understand what was done in FAST, nor what it tells us. I suggest to either extend these results massively or to leave it out completely.

To conclude, I believe the study in general is beneficial for the scientifc community. I expect the authors to use this as an initial paper with follow-ups with more technical content later on. Nevertheless, the authors shall add some novelty to this paper, such as suggested above, to justify a journal paper.

Please also note the supplement to this comment:
https://www.wind-energ-sci-discuss.net/wes-2020-45/wes-2020-45-RC2-supplement.pdf

**Supplement:**

[revised manuscript text omitted]

---

## Author Comment (AC2) · 6 May 2020

The authors thank the detailed and helpful comments. We have considered the feedback in the revised manuscript. Each comment is further addressed below.

- Comment 1: Section 3.1 needs more detail on the simple model as results are presented later. Some more specifics on thing such as how elements are modelled, what boundary conditions are used and what assumptions are made would be useful.

- R1: In order to better interpret the experimental results, a numerical model of the wind turbine was developed using Robot structural analysis software (Autodesk, 2016), following the technical drawings provided by the manufacturer. It is a simplified model, in which the operation of the turbine is not modelled. Rotational movement of the rotor and all control systems are disregarded, being the main purpose the simulation of the dynamic behaviour of the tower under the test conditions presented in the following section. It is considered that the foundation does not allow any kind of relative movements and is not considered the opening of the door (a specific numerical model for this detail has shown that it has a reduced influence on global behaviour). Thus, for the modelling of the tower was based on 3D bar elements to which the corresponding cross sections were assigned. Regarding blade modelling, at the time very detailed information was not available. Alternatively, starting from the NREL 5 MW reference wind turbine (Jonkman, Butterfield et al., 2009), the characteristics of the blades were scaled to be compatible with the wind turbine under study. The blades are modelled by 3D bar elements, divided into multiple sections to which the average mass, stiffness and inertia characteristics have been attributed. Since there is no rotation of the rotor, the blades were modelled with the pitch angle observed during the ambient vibration tests. The nacelle and hub are presented by concentrated loads applied at their centres of gravity. The connection between the tower, blades and the geometric centres of the nacelle and hub is modelled with rigid links of negligible mass.

Autodesk: Robot Structural Analysis Professional (Version 29.0.05650(x64)), 2016. Jonkman, J., Butterfield, S., Musial, W., and Scott, G.: Definition of a 5-MW Reference Wind Turbine for Offshore System Development: National Renewable Energy Laboratory (NREL), 2009.

- Comment 2: It's not clear how much data was used to construct the frequency tables in figure 8. Are these single observations of frequency or are they averages of multiple observations? It would also be good to know how much deviation is observed to give context to the level of difference between 'non-operational' and 'operational'.

- R2: The values of the natural frequencies presented in figure 8 were obtained from single observation (10 minutes time series of accelerations) under operating and nonoperating conditions. In the experimental campaigns conducted for a first estimation of the modal properties, several 10 minutes setups were measured, but in this paper only the values of one of the observations are presented. The number of datasets collected during the described ambient vibration tests is not enough for a reliable statistical characterization The evaluation of the variation of the modal parameters of the structure within the various operating regimes is still being performed.

- Comment 3: Line 129: it's mentioned that turbine 5 behaves 'differently' to the other turbines. Although a difference can be seen in figure 9 it would be clearer for the reader to say in the text what this difference is.

- R3: It is verified that the wind turbine 5 presents a different dynamic behaviour due to the differences observed at the values of the natural frequencies of the first and second tower bending modes associated with the side-side direction (1SS lower than the others and 2SS higher than the others).

- Comment 4: In section 4.2, I don't think it's ever mentioned what the sampling rate of any of the sensors are. Since a comparison is generally invited between the different sensors, such as that strain gauges can be used for an OMA purpose, it would be useful for the reader to know how comparable these sensors are regarding aspects such as the sampling rates.

- R4: Samples rates of:

Force-balance accelerometers = 20 Hz;

Strains and rotations tower monitoring systems = 50 Hz;

MEM accelerometers (blades and tower) = 62.5 Hz;

Blades strains monitoring system = 100 Hz

Some of these sampling rates resulted from hardware constrains. For the application of OMA algorithms, a sampling rate of 20Hz is already quite conservative taking into

account the natural frequencies of the most relevant modes.

- Comment 5: Line 242: It's mentioned that a model is conducted in FAST, also mentioned before in section 3.1, and the results are compared to the measurements. This is all moved on from too quickly. Why is a FAST model used? How important for that purpose is the difference from the measured results? Please give a bit more of the pertinent details on this and explain the aspects of this which might be of interest to the reader.

- R5: The main goal of the WindFarmSHM research project is the development, validation and optimization of a monitoring strategy to be applied at the level of the wind farm, suitable to both bottom fixed and floating solutions, which should be able to evaluate the structural condition of wind turbines and their consumed fatigue life. Since there are still very few floating wind turbines in operation and due to the confidentiality associated with this very promising technology, during the course of the project it is unlikely to have access to real monitoring data. Therefore, the development and validation of the monitoring strategy to be proposed for this type of offshore wind turbines will be based on artificial experimental data generated by numerical models. Firstly, numerical models of the instrumented onshore wind turbine in Tocha Wind Farm, taking into account their aerodynamics, control systems and flexibility of structural elements, are being constructed and tuned to replicate the experimental data. Then, these will be converted to floating wind turbine models including the hydrodynamics effects. The numerical models to be developed will also be used to simulate damage scenarios for both bottom fixed (e.g. stiffness reduction in the tower-foundation connection) and floating wind turbine (e.g. damage of a mooring line) to validate the algorithms that will be proposed for damage detection.

- Comment 6: Figure 1 is good for expressing the process, but the bottom three rows are confusing, what do the bars to the right of 'Blades', 'Tower' and 'Foundation' mean?

- R6: The bars represent in a simplistic fashion the damage detection check for the

structural elements (blades, tower and foundation) and the colour scale of the bar is related to the severity of the respective damage. The bars corresponding to the life-time prediction are related to the fatigue assessment of the structural elements (blades and tower) and indicate the percentage of useful life consumed up to the moment of analysis.

- Comment 7: Figure 9, it would help the reader to state in the caption that these measurements were from the non-operational condition.

- R7: Figure caption corrected for: Figure 9. Comparison of the natural frequencies of four wind turbines in non-operational condition (1st and 2nd pairs of bending modes).

- Comment 8: The description in the text of figure 17 (line 246) doesn't quite match the figure. It seems the results for 'Force-balance accelerometers' was added without updating the text.

- R8: Figure 17 shows the average spectra of the six force-balance accelerometers (first row), longitudinal deformations recorded by sensors A, B, C and D (second row) and clinometer 3 (third row), considering the rotor parked (left) and in operation (right).

- Comment 9: Line 122: please define the acronym ANPSD before using it.

- R9: ANPSD: average normalized auto-spectral density function

---

## Author Comment (AC3) · 13 May 2020

[revised manuscript text omitted]

---

## Author Response (AR1)

**Detailed point-by-point response to all referee comments**

**Title:** Development of new strategies for optimized structural monitoring of wind farms: description of the experimental field

**Author(s):** João Pacheco, Silvina Guimarães, Carlos Moutinho, Miguel Marques, José Carlos Matos and Filipe Magalhães

**MS No.:** wes-2020-45

**MS Type:** Research article

**Iteration:** Revised Submission

**Special Issue:** Wind Energy Science Conference 2019

**Introduction**

*First of all, the authors would like to acknowledge the work of the reviewers, which raised very pertinent questions and suggested corrections that have contributed to significantly improve the quality of the paper.*

*In order to allow an easier re-review, all the introduced modifications are explained in this document and are highlighted in the revised manuscript.*

**Interactive Comment 1**

Dmitri Tcherniak

dtcherniak@bksv.com

*Table 1 - Comment 1: page 2, line 47*

| Comment from Referee | p.2, l.47: "a few number of easy to install sensors" - " a few number" does not read good. |
|---|---|
| Author's response | This has been corrected in the revised manuscript. |
| Author's changes in manuscript | "In the project a very extensive instrumentation will be deployed in order evaluate different monitoring layout alternatives, but the final goal it to propose a minimal optimized monitoring layout based on ___reduced number of sensors that can be easily installed___." |

*Table 2 - Comment 2: page 4, line 68*

| Comment from Referee | p.4 l.68: the phrase "of variable height in section" is unclear same sentence;

p.4 l.68: "boltedconnections" – mistype |
|---|---|
| Author's response | This has been corrected in the revised manuscript. |
| Author's changes in manuscript | "The hub is placed at a height of 95 m and is supported by a steel tower, ___with a hollow circular cross-section with variable diameter and thickness, composed of four segments that are linked on site with bolted connections___." |

*Table 3 - Comment 3: page 4*

| | |
|---|---|
| **Comment from Referee** | p.4: Please check the consistency using a space between the value and the unit: "12m/s" but "20 m/s", also "100 m diameter rotor" but "height of 95m" |
| **Author's response** | In the revised document, it is now consistently used a space between the value and the unit. |
| **Author's changes in manuscript** | For example: 12 m/s, 95 m. |

*Table 4 - Comment 4: page 4, line 83*

| | |
|---|---|
| **Comment from Referee** | p.4, l.83: "since the main purpose is to simulate the dynamic behaviour of the tower". The main purpose of what? Of the entire project or of the numerical model? If the authors mean the entire project, it would be necessary to reflect this somewhere in the introduction. |
| **Author's response** | The main purpose of the numerical model is to simulate the dynamic behaviour of the tower. |
| **Author's changes in manuscript** | Section 3.1 was rewritten in the revised manuscript (see Table 9). |

*Table 5 - Comment 5: page 7, line 124*

| | |
|---|---|
| **Comment from Referee** | p.7, l.124 "the harmonic frequencies associated with to rotor operation" -> two rotor operations? |
| **Author's response** | The harmonic frequencies associated with the rotor operation $(\Omega, 3\Omega, 6\Omega, …)$. |
| **Author's changes in manuscript** | "The dashed vertical lines represent the harmonic frequencies associated with **_the_** rotor operation **_$(\Omega, 3\Omega, 6\Omega, …)$_**." |

*Table 6 - Comment 6: page 7, figure 8a*

| | |
|---|---|
| **Comment from Referee** | p.7, fig.8a: It is hard to see if there are "blue" peaks behind the red ones for the 1st and 2nd tower modes |
| **Author's response** | There is a blue peak for the 1st tower mode, but for the 2nd one there isn´t. Under non-operating conditions, the peak pairs associated with the first two tower mode pairs clearly stand out. In operating conditions, additional peaks associated with the rotor rotation frequency appear. The peaks associated with the second pair of bending modes become much more diffuse, which makes their tracking over time quite challenging. |
| **Author's changes in manuscript** | Paragraph included in the revised manuscript: "For the first tower bending modes there are two very pronounced peaks for the two considered operating conditions. For the second pair of tower bending modes only in non-operating conditions there is a clear peak in ANPSD. In figure 11 this comparison will be addressed again" |

*Table 7 - Comment 7: page 8, line 150*

| Comment from Referee | p.8, l.150: what is a MEM accelerometer? Do you mean MEMS accelerometers? |
|---|---|
| Author's response | It is a typo that was repeated several times, we meant MEMS (micro electromechanical systems). This is corrected in the revised manuscript. |
| Author's changes in manuscript | "Two alternative systems to characterize accelerations at the tower: a commercial system based on a set of very low noise accelerometers and a customized low-cost system based on MEMS *(micro electromechanical systems)* accelerometers designed and assembled in FEUP (Moutinho and Cunha, 2019);" |

*Table 8 - Comment 8: page 16, figure 20*

| Comment from Referee | p.16, fig.20. What are the units of the vertical axes? |
|---|---|
| Author's response | These are normalized power spectra, so without units. |
| Author's changes in manuscript | No changes have be made in the revised manuscript. |

Anonymous Referee #1

**Referee general comment:**

> "This paper presents the Tocha wind farm as well as the sensors installed and some initial results from those sensors. The paper is interesting, and it is useful to see the different types of results from different types of sensor. The purpose is as a precursor to future work, but I think the paper is interesting enough on its own merit. The paper is generally well structured and well put together, however I have several observations."

*Table 9 - Comment 1: page 4, section 3.1*

| | |
|---|---|
| **Comment from Referee** | Section 3.1 needs more detail on the simple model as results are presented later. Some more specifics on thing such as how elements are modelled, what boundary conditions are used and what assumptions are made would be useful. |
| **Author's response** | In order to better interpret the experimental results, a numerical model of the wind turbine was developed using ROBOT STRUCTURAL ANALYSIS software (Autodesk, 2016), following the technical drawings provided by the manufacturer. It is a simplified model, in which the operation of the turbine is not modelled. Rotational movement of the rotor and all control systems are disregarded, being the main purpose the simulation of the dynamic behaviour of the tower under the test conditions presented in the following section. |
| | It is considered that the foundation does not allow any kind of relative movements and is not considered the opening of the door (a specific numerical model for this detail has shown that it has a reduced influence on global behaviour). Thus, for the modelling of the tower was based on 3D bar elements to which the corresponding cross sections were assigned. |
| | Regarding blade modelling, at the time very detailed information was not available. Alternatively, starting from the NREL 5 MW reference wind turbine (Jonkman, Butterfield et al., 2009), the characteristics of the blades were scaled to be compatible with the wind turbine under study. The blades are modelled by 3D bar elements, divided into multiple sections to which the average mass, stiffness and inertia characteristics have been attributed. Since there is no rotation of the rotor, the blades were modelled with the pitch angle observed during the ambient vibration tests. |
| | The nacelle and hub are represented by concentrated loads applied at their centres of gravity. The connection between the tower, blades and the geometric centres of the nacelle and hub is modelled with rigid links of negligible mass. |
| | Autodesk: Robot Structural Analysis Professional (Version 29.0.05650(x64)), 2016. |
| | Jonkman, J., Butterfield, S., Musial, W., and Scott, G.: Definition of a 5-MW Reference Wind Turbine for Offshore System Development: National Renewable Energy Laboratory (NREL), 2009. |
| **Author's changes in manuscript** | Section 3.1 was rewritten in the revised manuscript based on the description presented above. |

*Table 10 - Comment 2: page 7, figure 8*

| | |
|---|---|
| *Comment from Referee* | It's not clear how much data was used to construct the frequency tables in figure 8. Are these single observations of frequency or are they averages of multiple observations? It would also be good to know how much deviation is observed to give context to the level of difference between 'non-operational' and 'operational'. |
| *Author's response* | The values of the natural frequencies presented in figure 8 were obtained from single observations (10 minutes time series of accelerations) under operating and non-operating conditions. In the experimental campaigns conducted for a first estimation of the modal properties, several 10 minutes setups were measured, but in this paper only the values of one of the observations are presented. The number of datasets collected during the described ambient vibration tests is not enough for a reliable statistical characterization The evaluation of the variation of the modal parameters of the structure within the various operating regimes is still being performed. |
| *Author's changes in manuscript* | Add line 93 of the revised manuscript: "**_Several 10 minutes of accelerations time series_** were measured **_(sample rate of 100 Hz)_** with 4 standalone seismographs (Figure 5 a), with internal tri-axial force balance sensors, that were placed in the horizontal platforms of the tower (Figure 5 b). Figure 8 caption on revised manuscript corrected for: "Figure 8. Ambient Vibration test results for wind turbine 1 in operating and non-operating conditions: average power spectra and natural frequencies **_(results obtained from 10 minutes single observation setups with very low variance of the environment and operational parameters)_**." |

*Table 11 - Comment 3: page 7, line 129*

| | |
|---|---|
| *Comment from Referee* | Line 129: it's mentioned that turbine 5 behaves 'differently' to the other turbines. Although a difference can be seen in figure 9 it would be clearer for the reader to say in the text what this difference is. |
| *Author's response* | It is verified that wind turbine 5 presents a different dynamic behaviour due to the differences observed at the values of the natural frequencies of the first and second tower bending modes associated with the side-side direction (1SS lower than the others and 2SS higher than the others). |
| *Author's changes in manuscript* | "All of them present quite similar natural frequencies, but wind turbine 5 seams to present a slightly different behaviour, **_expressed by the differences observed at the values of the natural frequencies of the first and second tower bending modes associated with the side-side direction (1SS lower than the others and 2SS higher than the others)_**." |

*Table 12 - Comment 4: section 4.2*

| Comment from Referee | In section 4.2, I don't think it's ever mentioned what the sampling rate of any of the sensors are. Since a comparison is generally invited between the different sensors, such as that strain gauges can be used for an OMA purpose, it would be useful for the reader to know how comparable these sensors are regarding aspects such as the sampling rates. |
|---|---|
| *Author's response* | Samples rates of:
• force-balance accelerometers = 20 Hz;
• Strains and rotations tower monitoring systems = 50 Hz;
• MEM accelerometers (blades and tower) = 62.5 Hz;
• Blades strains monitoring system = 100 Hz
Some of these sampling rates resulted from hardware constrains. For the application of OMA algorithms, a sampling rate of 20Hz is already quite conservative taking into account the natural frequencies of the most relevant modes. |
| *Author's changes in manuscript* | The sampling frequencies for each of the described monitoring systems have been added in the respective sections of the revised manuscript. |

*Table 13 - Comment 5: page 13, line 242*

| Comment from Referee | Line 242: It's mentioned that a model is conducted in FAST, also mentioned before in section 3.1, and the results are compared to the measurements. This is all moved on from too quickly. Why is a FAST model used? How important for that purpose is the difference from the measured results? Please give a bit more of the pertinent details on this and explain the aspects of this which might be of interest to the reader. |
|---|---|
| *Author's response* | The main goal of the WindFarmSHM research project is the development, validation and optimization of a monitoring strategy to be applied at the level of the wind farm, suitable to both bottom fixed and floating solutions, which should be able to evaluate the structural condition of wind turbines and their consumed fatigue life.

Since there are still very few floating wind turbines in operation and due to the confidentiality associated with this very promising technology, during the course of the project it is unlikely to have access to real monitoring data. Therefore, the development and validation of the monitoring strategy to be proposed for this type of offshore wind turbines will be based on artificial experimental data generated by numerical models. Firstly, numerical models of the instrumented onshore wind turbine in Tocha Wind Farm, taking into account their aerodynamics, control systems and flexibility of structural elements, are being constructed and tuned to replicate the experimental data. Then, these will be converted to floating wind turbine models including the hydrodynamics effects.

The numerical models to be developed will also be used to simulate damage scenarios for both bottom fixed (e.g. stiffness reduction in the tower-foundation connection) and floating wind turbine (e.g. damage of a mooring line) to validate the algorithms that will be proposed for damage detection. |
| *Author's changes in manuscript* | In Table 9, it is included a suggestion for an update on the text that introduces the FAST model. Since the construction of this model is out of scope of the present paper, the authors believe that it is not necessary to provide further details, the interested reader is referred to another paper on this specific topic (Pimenta, Branco et al., 2019). |

*Table 14 - Comment 6: page 2, figure 1*

| | |
|---|---|
| **Comment from Referee** | Figure 1 is good for expressing the process, but the bottom three rows are confusing, what do the bars to the right of 'Blades', 'Tower' and 'Foundation' mean? |
| **Author's response** | The bars represent in a simplistic fashion the damage detection check for the structural elements (blades, tower and foundation) and the colour scale of the bar is related to the severity of the respective damage. The bars corresponding to the lifetime prediction are related to the fatigue assessment of the structural elements (blades and tower) and indicate the percentage of useful life consumed up to the moment of analysis. |
| **Author's changes in manuscript** | Figure caption corrected for:
"Figure 1. Monitoring strategy. ***Meaning of the horizontal bars at the bottom: the colour bars represent in a simplistic fashion the structural health, the green/white bars represent the consumed fatigue life***." |

*Table 15 - Comment 7: page 7, figure 9*

| | |
|---|---|
| **Comment from Referee** | Figure 9, it would help the reader to state in the caption that these measurements were from the non-operational condition. |
| **Author's response** | This has been corrected in the revised manuscript. |
| **Author's changes in manuscript** | Figure caption corrected for:

"Figure 9. Comparison of the natural frequencies of four wind turbines ***in non-operational conditions*** (1st and 2nd pairs of bending modes)." |

*Table 16 - Comment 8: page 13, line 246*

| | |
|---|---|
| **Comment from Referee** | The description in the text of figure 17 (line 246) doesn't quite match the figure. It seems the results for 'Force-balance accelerometers' was added without updating the text. |
| **Author's response** | This has been corrected in the revised manuscript. |
| **Author's changes in manuscript** | "Figure 17 shows the average spectra of ***the six force-balance accelerometers (first row)***, 4 longitudinal deformations recorded by sensors A, B, C and D (***second*** row) and clinometer 3 (***third*** row), considering the rotor parked (left) and in operation (right)." |

*Table 17 - Comment 9: page 7, line 122*

| | |
|---|---|
| **Comment from Referee** | Line 122: please define the acronym ANPSD before using it. |
| **Author's response** | ANPSD: average normalized auto-spectral density function. This has been included in the revised manuscript. |
| **Author's changes in manuscript** | "Thus, Figure 8 compares ANPSD ***(average normalized auto-spectral density function)*** obtained for stopped rotor and in operation." |

*Table 18 - Comment 10*

| | |
|---|---|
| *Comment from Referee* | A few minor typos and some grammatical errors throughout, though not too bad. Some examples:

1. I think you should capitalize 'Robot Structural Model' (page 4, line 80);

2. Check grammar in line 86-87, page 4.

3. Table in Figure 7 (page 6): I think the second $f_{exp}$ should be $f_{model}$

4. Please check the grammar in the sentence at line 144, page 8: "The main goal of the simple monitoring layout is to characterize the differences in the behaviour of wind turbines and to understand the interaction between neighbouring wind turbines." |
| *Author's response* | These typos have been corrected in the revised manuscript. Furthermore, the revised manuscript has been carefully read and some other typos corrected. |
| *Author's changes in manuscript* | 1. "In order to understand and interpret the experimental results, a numerical model of the wind turbine was developed using **_ROBOT STRUCTURAL ANALYSIS_** software (Autodesk, 2016), according to the technical drawings provided by the manufacturer."

2. Section 3.1 was rewritten in the revised manuscript (see Table 9)

3. Second row: $f_{model}\ [Hz]$

4. This sentence has been rewritten in the revised manuscript: "The simple monitoring layout has two main objectives: to characterize and identify differences in the dynamic behavior of wind turbines and to understand the interaction of the wake effects between nearby wind turbines." |

**Interactive Comment 3**

Lisa Ziegler (Referee)

l.ziegler@enbw.com

**Referee general comment:**

*"The authors present their experimental field for structural monitoring of onshore wind turbines. They introduce sensor setups and first results on modal parameters. The topic has high relevance for the wind industry due to aging fleet of assets.*

*The paper is cleary written, the content is sound and well presented.*

*Introduction misses a review on state-of-art and existing literature. What is the research gap you wish to fill?*

*Presented results are clearly, however, I miss novelty here. Furthermore, I wish there would be critical discussion in the paper. For example, interesting questions would be:*

*\* Why is the specific instrumentation chosen?*

*\* How are number and positions of sensors chosen, e.g. sensitivity study of desired results to sensor palcement?*

*\* How do you deal with measurement noise and varying operational conditions?*

*\* How do you clear and pre-process data?*

*In addition the following is missing or must be adapted:*

*\* Results on the comparison between bending moments obtained from strain gauges and clinometers shall be presented.*

*\* A feedback from results of blade monitoring to tower monitoring. Can you now explain some more of the excitation frequencies?*

*\* Blade results are presented although the calibration is not completed. Please finish first the calibration, then present results.*

*I do not understand why the results from FAST are presented. There are not enough details given to understand what was done in FAST, nor what it tells us. I suggest to either extend these results massively or to leave it out completely.*

*To conclude, I believe the study in general is beneficial for the scientifc community. I expect the authors to use this as an initial paper with follow-ups with more technical content later on. Nevertheless, the authors shall add some novelty to this paper, such as suggested above, to justify a journal paper.."*

**Author's response:**

The authors acknowledge the very detailed analysis of the paper that contributed to the implementation of significant improvements in the revised manuscript.

The main goal of the paper is indeed to present for the first time in a journal paper the ongoing experimental campaign. The detailed data processing of each monitoring component is an ongoing work that will for sure lead to complementary publications.

The authors' feedback to more detailed comments are presented in the next tables.

*Table 19 - Comment 1: page 1, abstract*

| Comment from Referee | Abstract misses an overview of (quantitative) results. What was achieved in the paper? |
|---|---|
| Author's response | We assumed as important goals of the paper the demonstration of the monitoring system good performance and the presentation of preliminary results, as stated in the last sentence of the abstract. |
| Author's changes in manuscript | The last sentence of the abstract was improved in order to include a more complete description of the results presented in the paper:

"At this preliminary stage, the capabilities of the very extensive monitoring layout will be demonstrated. The results presented in paper demonstrate the ability of the different monitoring components to track the modal parameters of the system composed by tower and rotor and to characterize the internal loads at the tower base and blade roots." |

*Table 20 - Comment 2: page 2*

| Comment from Referee | A review of current state-of-the art is missing. What has already been published for structural monitoring for wind farms? Where is your research gap? |
|---|---|
| Author's response | The article is already quite extensive and for this reason it was decided not to include a very complete review of current state-of-the art. Still, the authors understand the concern of the reviewer and so, some new references were added in the revised manuscript.

Weijtjens, W., Noppe, N., Verbelen, T., Iliopoulos, A., and Devriendt, C.: Offshore wind turbine foundation monitoring, extrapolating fatigue measurements from fleet leaders to the entire wind farm. Journal of Physics: Conference Series, 753(9), 1742-6596, 2016.

Loraux, C. and Brühwiler, E.: The use of long term monitoring data for the extension of the service duration of existing wind turbine support structures. Journal of Physics: Conference Series, 753(7), 1742-6596, 2016.

Weijtjens, W., Verbelen, T., Capello, E and Devriendt, C.: Vibration based structural health monitoring of the substructures of five offshore wind turbines. Procedia Engineering, 199, 2017.

Considering the actual state-of-the art, our research is motivated by the need to optimize the monitoring systems for cost reduction, the need for an approach at the level of the wind farm, based on the instrumentation of only few wind turbines and the need to demonstrate the advantages of the proposed monitoring tools in the context of innovative floating wind turbines concepts. |
| Author's changes in manuscript | Line 33 on the revised manuscript was improved in order to include more complete information.

"Considering this background ***and previous research (Weijtjens, Noppe et al., 2016, Lorax and Brühwiler, 2016, Weijtjens, Verbelen, et al. 2017)*** the main goal of the WindFarmSHM research project is the development, validation and optimization of new methodologies to continuously assess the structural elements of wind turbines: tower, blades and 35 foundation." |

*Table 21 - Comment 3: page 2, line 35*

| | |
|---|---|
| **Comment from Referee** | "adequate for onshore and offshore solutions"

Onshore and floating? I somehow miss the transition. What about offshore bottom-fixed? |
| **Author's response** | The main goal is to develop and apply the algorithms developed to all types of wind farms (onshore, bottom-fixed and floating). In Portugal, there are more than 2500 onshore wind turbines in operation and a floating offshore wind farm that is currently under construction (WindFloat Atlantic). For these reasons, we intend to develop tools for onshore and floating in the first place. |
| **Author's changes in manuscript** | "The monitoring strategy is being designed to be applied in the context of a wind farm, adequate for onshore and floating solutions ***(the two types of foundation being used in Portugal),*** using optimized instrumentation layouts at a subgroup of wind turbines, and taking profit from the data provided by the acquisition systems already available in all wind turbines (SCADA), for the use of extrapolation techniques to assess all the wind turbines of the same wind farm (Figure 1)." |

*Table 22 - Comment 4: page 2, line 40*

| | |
|---|---|
| **Comment from Referee** | "artificial experimental data"

There are no measurements on floating platforms, right? I believe "experimental data" might be misleading. |
| **Author's response** | Since there are still very few floating wind turbines in operation and due to the confidentiality associated with this very promising technology, during the course of the project it is unlikely to have access to real monitoring data. Therefore, the development and validation of the monitoring strategy to be proposed for this type of offshore wind turbines will be based on artificial experimental data generated by numerical models. |
| **Author's changes in manuscript** | "The research project will include three monitoring layouts of wind turbines of an onshore wind farm, comprehending accelerometers, strain gages and clinometers and the development of numerical models for the generation of ***virtual monitoring*** data to validate the monitoring strategy in floating wind turbines." |

*Table 23 - Comment 5: page 2, line 44*

| Comment from Referee | "detection of stiffness reductions motivated by the appearance of damage"

What damages do you plan to detect here? |
|---|---|
| Author's response | As already performed in previous works, The experimentally identified natural frequencies will be modified with natural frequency shifts associated with the simulated damages, such as:

• Scour problems at the foundation of an offshore monopile wind turbine;
• Foundation problems in onshore wind turbines;
• Blade damage;
• Mooring line problems in floating wind turbines; |
| Author's changes in manuscript | "The data processing will be based on the continuous evaluation of the parameters that drive the structure dynamic behaviour (vibration frequencies and damping) estimated from the structure response to ambient excitation (wind, waves, currents, soil vibrations) and advanced statistical modelling, having in mind two main goals: detection of stiffness reductions motivated by the appearance of damage *(as performed in (Oliveira, Magalhães, et al. 2018))* and evaluation of the remaining fatigue life of the main structural components (Figure 1)."

Oliveira, G., Magalhães, F. Cunha, A. and Caetano, E.: Vibration based damage detection in a wind turbine using one year of data. Structural Control and Health Monitoring, 25, 11, 2018. [https://doi.org/10.1002/stc.2238]. |

*Table 24 - Comment 6: page 2, line 45*

| Comment from Referee | "remaining fatigue life"

How do you plan to calculate the remaining fatigue life? Do you have all design information needed for this? |
|---|---|
| Author's response | We are developing algorithms for data processing that should permit the evaluation of the remaining fatigue life of the main structural components based on the direct measurement of the curvatures with strain gages, curvature measurements using pairs of clinometers and accelerations to support the application of a virtual sensors approach.

We have the necessary details for the tower and blades, but this is an ongoing research that it out of scope of the present paper |
| Author's changes in manuscript | No changes have been made in the revised manuscript. |

*Table 25 - Comment 7: page 2, line 47*

| Comment from Referee | "it" |
|---|---|
| Author's response | This has been corrected in the revised manuscript. |
| Author's changes in manuscript | "In the project a very extensive instrumentation is being deployed in order evaluate different monitoring layout alternatives, but the final goal *is* to propose a minimal optimized monitoring layout based on a few number of easy to install sensors." |

*Table 26 - Comment 8: page 2, figure 1*

| | |
|---|---|
| **Comment from Referee** | Very nice figure, gives a good overview.

"Damage" Detection (spelling) |
| **Author's response** | This has been corrected in the revised manuscript. |
| **Author's changes in manuscript** | "Damage Detection" |

*Table 27 - Comment 9: page 3, line 56*

| | |
|---|---|
| **Comment from Referee** | Would be nice to see a drawing or picture of this, if available. |
| **Author's response** | Due to confidentiality agreements, we are not allowed to present constructive details of the wind turbines. |
| **Author's changes in manuscript** | No changes have been made in the revised manuscript. |

*Table 28 - Comment 10: page 4, line 68*

| | |
|---|---|
| **Comment from Referee** | "boltedconnections" |
| **Author's response** | This has been corrected in the revised manuscript. |
| **Author's changes in manuscript** | "The hub is placed at a height of 95 m and is supported by a steel tower, ***with a hollow circular cross-section with variable diameter and thickness, composed of four segments that are connected by bolted connections***." |

*Table 29 - Comment 11: page 4, line 86*

| | |
|---|---|
| **Comment from Referee** | "has also important"

What does this mean? Do you use the modal parameters to tune the FAST model? |
| **Author's response** | We used in the FAST model the mode shapes of the tower and blades obtained with the finite element model and also the rotational stiffness of the tower foundation. The procedure is explained the reference (Pimenta, Branco et al., 2019). |
| **Author's changes in manuscript** | In the revised manuscript the last paragraph of section 3.1 was rephrased:

"Still, it's important to note that more advanced models are currently being developed in FAST (Sprague, Jonkman et al., 2015) using some structural information that was derived from the previously described model. All the details of the FAST model are presented in (Pimenta, Branco et al., 2019)." |

*Table 30 - Comment 12: page 5, line 90*

| | |
|---|---|
| **Comment from Referee** | "divide into" |
| **Author's response** | This has been corrected in the revised manuscript. |
| **Author's changes in manuscript** | "The set of ambient vibration tests was ***divided into*** two campaign." |

*Table 31 - Comment 13: page 5, figure 5*

| | |
|---|---|
| **Comment from Referee** | Nice presentation of setup. However, some important technical details pn the measurement concept are missing: How long did you measure? What resolution? Did you monitor SCADA time-synchronized? |
| **Author's response** | In the experimental campaigns conducted for a first estimation of the modal properties, several 10 minutes time series of accelerations under operating and non-operating conditions were measured (sample rate: 100 Hz).

In this study, the owner of the wind farm provides SCADA data with two types of sample: records the mean, maximum and minimum value from 10 min period (SCADA 10min) and data with a sampling interval of 15 sec (SCADA high resolution).

Some further details were included in the revised manuscript. |
| **Author's changes in manuscript** | Add line 93 of the revised manuscript:
"***Several 10 minutes of accelerations time series*** were measured ***(sample rate of 100 Hz)*** with 4 standalone seismographs (Figure 5 a), with internal tri-axial force balance sensors, that were placed in the horizontal platforms of the tower (Figure 5 b).

Add line 100 of the revised manuscript:
"***The owner of the wind farm provides SCADA data with the mean, maximum and minimum value from 10 minutes period, important information for the accelerations processing***" |

*Table 32 - Comment 14: page 5, line 107*

| | |
|---|---|
| **Comment from Referee** | "Among the various peaks identified are two that clearly stand out: one near 0.25 Hz and another near 1.80 Hz."

Why do these stand out? What is the difference to other frequencies with stable poles and good MAC, e.g. around 2.2 Hz or 4.5 Hz? |
| **Author's response** | The peaks around 0.25 Hz and 1.80 Hz have high values of the average power spectra for both main directions (FA and SS).

As opposition to the power spectra, in the stabilization diagram we cannot identify the energy that is associated with each alignment of stable poles. The referred stable poles have less energy in the tower response, so they are probably associated with rotor modes. The numerical model also helped in the selection of the frequencies that are associated with tower modes. |
| **Author's changes in manuscript** | In the revised manuscript the line 107 was rephrased:

"Among the various peaks identified, there are two that clearly stand out in the presented ANPSD (higher amplitudes): one near 0.25 Hz and another near 1.80 Hz." |

*Table 33 - Comment 15: page 5, line 108*

| Comment from Referee | "These peaks correspond to the first and second pairs of tower bending modes." How do you confirm this? |
|---|---|
| Author's response | Comparing the experimental results with the numerical ones in the figure 7. |
| Author's changes in manuscript | "***Comparing the experimental results with numerical ones (figure 7), it is confirmed that*** these peaks correspond to the first and second pairs of tower bending modes." |

*Table 34 - Comment 16: page 6, figure 6*

| Comment from Referee | Please improve the caption and layout. Use a/b/c/d or top left/ right, bottom, ..., to indicate what we see where. Why are in the top right figure 3 red boxed but only two zooms below? (explanation in the caption needed) |
|---|---|
| Author's response | For the first two red boxes the two vertical alignments (FA and SS) of the poles in the stabilization diagrams are very close and barely noticeable in the figure top right. The two alignments for the third box are clearly visible in the top right figure, so there was no need to present a zoom. In the revised manuscript the figure caption was changed considering your suggestions. |
| Author's changes in manuscript | Figure 6 caption: "Figure 6. Ambient Vibration test results for wind turbine 1: ***a)*** average power spectra for FA and SS directions; ***b)*** stabilization diagram produced by the SSI-COV method; ***c)*** two zooms of this diagram ***for first (left) and second (right) pairs of tower bending modes***." |

*Table 35 - Comment 17: page 6, figure 7*

| Comment from Referee | 2x fexp? |
|---|---|
| Author's response | "fmodel" on second row. |
| Author's changes in manuscript | $f_{model}$ |

*Table 36 - Comment 18: page 7, line 124*

| Comment from Referee | "harmonic frequencies" To which rotor speed correspond these harmonics? Is the wind turbine already at rated speed at 10m/s? How much variance did you have in the rotor speed during your test campaign? |
|---|---|
| Author's response | These harmonics correspond to (data in the table below the figure on the left) :
 - rotor speed = 14.9 rpm;
 - wind speed = 11.3 m/s;
For these results we selected a 10 minutes time series of accelerations with very low variance of the environment an operational parameters. |
| Author's changes in manuscript | "The dashed vertical lines represent the harmonic frequencies associated with to rotor operation. The results obtained in terms of natural frequencies ($f$) are also compared for the identified vibration modes for the two analysed situations ***(environment and operational parameters shown at the table on bottom left)***."

Figure 8 caption on revised manuscript corrected for:
"Figure 8. Ambient Vibration test results for wind turbine 1 in operating and non-operating conditions: average power spectra and natural frequencies ***(results obtained from 10 minutes single observations setups with very low variance of the environment and operational parameters)***." |

*Table 37 - Comment 19: page 7, figure 8 (left)*

| Comment from Referee | Do I see it correctly that the first fnat is almost identical to 1P? Typically, this shall be avoided in design (resonance, high loads). Are you sure about these results? |
|---|---|
| Author's response | Although the first natural frequency is very close to the first harmonic (for the particular operating conditions associated with the presented plot), these frequencies are far enough apart. The control mechanisms of the wind turbine prevent these two frequencies from being too close. |
| Author's changes in manuscript | We prefer to not included any comment on this, because it is a sensitive topic for the wind turbine manufacture |

*Table 38 - Comment 20: page 7, figure 8 (right)*

| Comment from Referee | How do you explain the difference in frequencies between operation and non-operation? |
|---|---|
| Author's response | It is easily understood that the modal parameters of the wind turbine change due to different environmental and operational conditions, this is described for instance in reference (Oliveira, 2018). The evaluation of the variation of the modal parameters of the structure between within the various operating regimes is still being performed. |
| Author's changes in manuscript | No changes have been made in the revised manuscript. |

*Table 39 - Comment 21: page 6, figure 7*

| | |
|---|---|
| **Comment from Referee** | "present a slightly different behaviour"

Do you know where this comes from? Are tower and foundation properties identical? |
| **Author's response** | All wind turbines have the same physical and geometrical characteristics, so in theory they should have a similar dynamic behaviour. The different dynamic behaviour of turbine 5 might be explained by the fact that it is subject to important wake effects, but this a topic that deserves further studies. |
| **Author's changes in manuscript** | Since we do not have definite justification for the observed differences, we prefer not to include comments on this in the revised manuscript. |

*Table 40 - Comment 22: page 8, line 135*

| | |
|---|---|
| **Comment from Referee** | "dynamic behavior of all generators"

Misleading formulation. You are not interested in the dynamics of the generator in the drive train. |
| **Author's response** | This has been corrected in the revised manuscript. |
| **Author's changes in manuscript** | "In order to obtain data representative of the dynamic behavior of all **_wind turbines_** and based on the results of the ambient vibration tests described above, the experimental campaign includes the following three instrumentation layouts:" |

*Table 41 - Comment 23: page 8, line 137*

| | |
|---|---|
| **Comment from Referee** | "very complete"

Extended? |
| **Author's response** | This has been corrected in the revised manuscript. |
| **Author's changes in manuscript** | "An **_extended_** monitoring layout installed on wind turbine 1;" |

*Table 42 - Comment 24: page 8, line 140*

| | |
|---|---|
| **Comment from Referee** | "Based on the wind conditions of the site (Figure 3) and the position of each wind turbine in the wind farm (Figure 2) wind turbine 5 is the wind turbine where higher turbulence is expected because it is in the wake of the other wind turbines, while wind turbine 1 is exposed to less disturbed winds."

Repetition |
| **Author's response** | This has been corrected in the revised manuscript. |
| **Author's changes in manuscript** | See Table 43. |

*Table 43 - Comment 25: page 8, line 142*

| | |
|---|---|
| **Comment from Referee** | "For this reason, wind turbine 1 was instrumented according to the complete layout, while the intermediate layout was applied in wind turbine 5."

Why this? Is it not even more interesting to completely instrument a turbine which sees both: free stream and wake conditions? In order to extrapolate results it should be beneficial if as many conditions as possible are represented in the dataset. |
| **Author's response** | In fact depending on the wind direction all the wind turbines of the farm are subjected to free stream and wake conditions. Considering the predominant wind direction WT 5 is more frequently affected by wakes. |
| **Author's changes in manuscript** | In the revised manuscript the paragraph of the line 140 was rephrased:

"The distribution of the alternative monitoring layouts in the wind turbines of the farm was conditioned by the available time slot for installation of equipment (usually scheduled during other maintenance operations) and our will to instrument the rotor of one wind turbine that for the predominance wind direction (north) is loaded by an unperturbed flow and another one the is influenced by the wakes of the other turbines (see Figure 2 and 3)." |

*Table 44 - Comment 26: page 8, line 162*

| | |
|---|---|
| **Comment from Referee** | "SCADA system"

What resolution? Do you use 10-min statistics or 1Hz data of SCADA? |
| **Author's response** | In this study, the owner of the wind farm provides SCADA data with two types of sample:

- SCADA records the mean, maximum and minimum value from 10 min period (SCADA 10min);

- SCADA data with a sampling interval of 15 sec (SCADA high resolution). |
| **Author's changes in manuscript** | Add line 162:

"It should be noted that data on the environmental and operational conditions of each wind turbine is being obtained through the SCADA system *(10 minutes averages and sampled at 15 seconds)*." |

*Table 45 - Comment 27: page 9, line 174*

| | |
|---|---|
| **Comment from Referee** | How do you choose the placement of the accelerometers? Why 6 pieces? |
| **Author's response** | These 3 sections coincide with the height of the technical platforms, in order to facilitate the installation and maintenance of the sensors.

Instrumentation of three sections of the tower along two orthogonal horizontal directions (unidirectional accelerometers). |
| **Author's changes in manuscript** | Line 173:
"As depicted in Figure 10, this involved the instrumentation of 3 sections of the tower along two orthogonal horizontal directions. *These 3 sections coincide with the height of the technical platforms, in order to facilitate the installation and maintenance of the monitoring equipment.*"

Line 172:
"In order to obtain the best possible characterization of the tower accelerations a commercial system based on 6 force-balance *unidirectional* accelerometers connected to a 24bits acquisition system was deployed." |

*Table 46 - Comment 28: page 8, line 179*

| | |
|---|---|
| **Comment from Referee** | "MEM based system"

What resolution do you measure? |
| **Author's response** | Samples rates of:
• force-balance accelerometers = 20 Hz;
• Strains and rotations tower monitoring systems = 50 Hz;
• MEM accelerometers (blades and tower) = 62.5 Hz;
• Blades strains monitoring system = 100 Hz
Some of these sampling rates resulted from hardware constrains. For the application of OMA algorithms, a sampling rate of 20Hz is already quite conservative taking into account the natural frequencies of the most relevant modes. |
| **Author's changes in manuscript** | The sampling frequencies for each of the described monitoring systems will be added in the respective section. |

*Table 47 - Comment 29: page 11, line 204*

| | |
|---|---|
| **Comment from Referee** | "on important" |
| **Author's response** | This has been corrected in the revised manuscript. |
| **Author's changes in manuscript** | "These monitoring components are essential for fatigue assessment of the tower and **_one_** important goal is the evaluation of two alternatives for estimating static and dynamic bending moment diagrams along the tower" |

*Table 48 - Comment 30: page 11, line 205 and page 13, line 251*

| | |
|---|---|
| **Comment from Referee** | "extensions measurements" and "extensions"

What is this? Strain measurements? |
| **Author's response** | This has been corrected in the revised manuscript. |
| **Author's changes in manuscript** | "…using **_strain_** and rotation measurements, combined with accelerometers."

"…so measuring **_strains_** can be very useful in distinguishing tower modes from the rotor modes observed in the tower. |

*Table 49 - Comment 31: page 11, line 216*

| | |
|---|---|
| **Comment from Referee** | How do you measure bending moments from the clinometers? |
| **Author's response** | By measuring the rotation in two sections of the tower ($rot_A$ $and$ $rot_B$) it is possible to determine the curvature of the middle section($y''_{AB}$). $$y''_{AB} = \frac{rot_A - rot_B}{length\ \overline{AB}}$$ $$Bending\ Moments = E * I * y''_{AB}$$ $E: Young\ modulus\ of\ the\ steel\ of\ the\ tower$ $I: Second\ moment\ of\ inertia\ of\ the\ middle\ cross\ section\ AB$ |
| **Author's changes in manuscript** | No changes have be made in the revised manuscript. |

*Table 50 - Comment 32: page 12, figure 14*

| | |
|---|---|
| **Comment from Referee** | This strain gauge seems quite close to a welded joint. Have you checked that yours are out of range of stress concentration due to the weld? |
| **Author's response** | The monitoring project was prepared according to code IEC 61400-13. There is no influence of welding on measurements. |
| **Author's changes in manuscript** | No changes have be made in the revised manuscript. |

*Table 51 - Comment 33: page 13, line 239*

| | |
|---|---|
| **Comment from Referee** | Can you please present the bending moments you obtain with the SGs in comparison to the clinometers? |
| **Author's response** | The results obtained with the clinometers still need further calibration. In theory equivalent values should be obtained, but we are observing some differences that triggered further studies. |
| **Author's changes in manuscript** | No changes have be made in the revised manuscript. |

*Table 52 - Comment 34: page 13, figure 16*

| | |
|---|---|
| **Comment from Referee** | Where do the differences between FAST and the measurements come from? Do you use data from the met mast to calibrate the inflow in field? |
| | I am not sure what this comparison shall tell us. That the FAST results show some similarities in the time series on one side, are quite different on the other side? E.g. Side-side moments are considerably different in the left figure (much larger amplitudes in measurements). |
| **Author's response** | The goal of the plot is to provide a qualitative comparison. The inflow is different, we just generated a wind with the same average speed and turbulence. |
| | More extensive comparisons between numerical and experimental results are being performed. |
| **Author's changes in manuscript** | In the revised manuscript the line 242 was rephrased: |
| | "The experimental results are compared with numerical ones, obtained from a model developed on FAST and calibrated using the methodology described in (Pimenta, Branco et al., 2019). Please note this is just a qualitative comparison, the inflows in the experiment and numerical model are difference, only the average wind speed and turbulence intensity are the same." |

*Table 53 - Comment 35: page 13, figure 16*

| | |
|---|---|
| **Comment from Referee** | FAST (capital letters) |
| **Author's response** | This has benn corrected in the revised manuscript. |
| **Author's changes in manuscript** | FAST |

*Table 54 - Comment 36: page 13, line 251*

| | |
|---|---|
| **Comment from Referee** | "more pronounced and clearer" |
| | Why is this the case? |
| **Author's response** | In the average spectra of longitudinal strains gauge, the peaks motivated by the tower bending modes are clearer than in the spectrum obtained from accelerations. The tower strains measurements are also less influenced by the rotor modes. |
| **Author's changes in manuscript** | No changes have been made in the revised manuscript. |

*Table 55 - Comment 37: page 16, figure 20*

| Comment from Referee | How can you feedback this to the tower monitoring? Can you now explain more of the excitation frequencies you see in the tower? |
|---|---|
| Author's response | Indeed one of the goals of this monitoring component is a more deep understanding of the tower measurement. However, this is still ongoing research. |
| Author's changes in manuscript | No changes have been made in the revised manuscript. |

*Table 56 - Comment 38: page 17, line 303*

| Comment from Referee | I believe that these results shall not be presented if they are not completed yet. Please finish first the calibration and then present the results. |
|---|---|
| Author's response | The higher inaccuracies are in the lead-lag direction, so the second plot was eliminated in the revised manuscript. |
| Author's changes in manuscript | The second plot (below) was eliminated in the revised manuscript. |

*Table 57 - Comment 39: page 18, figure 24*

[revised manuscript text omitted]